# Counterfactual Generation with Identifiability Guarantees

**Hanqi Yan**[1,4]*, **Lingjing Kong**[2]*, **Lin Gui**[3], **Yuejie Chi**[2], **Eric Xing**[2,4], **Yulan He**[1,3], **Kun Zhang**[2,4]

[1]University of Warwick, [2]Carnegie Mellon University,
[3]King's College London, [4]Mohamed Bin Zayed University of Artificial Intelligence

## Abstract

Counterfactual generation lies at the core of various machine learning tasks, including image translation and controllable text generation. This generation process usually requires the identification of the disentangled latent representations, such as content and style, that underlie the observed data. However, it becomes more challenging when faced with a scarcity of paired data and labeling information. Existing disentangled methods crucially rely on oversimplified assumptions, such as assuming independent content and style variables, to identify the latent variables, even though such assumptions may not hold for complex data distributions. For instance, food reviews tend to involve words like "*tasty*", whereas movie reviews commonly contain words such as "*thrilling*" for the same positive sentiment. This problem is exacerbated when data are sampled from multiple domains since the dependence between content and style may vary significantly over domains. In this work, we tackle the domain-varying dependence between the content and the style variables inherent in the counterfactual generation task. We provide identification guarantees for such latent-variable models by leveraging the relative sparsity of the influences from different latent variables. Our theoretical insights enable the development of a do**M**ain **A**dap**T**ive coun**T**erfactual g**E**neration model, called (**MATTE**). Our theoretically grounded framework achieves state-of-the-art performance in unsupervised style transfer tasks, where neither paired data nor style labels are utilized, across four large-scale datasets.

## 1 Introduction

Counterfactual generation serves as a crucial component in various machine learning applications, such as controllable text generation and image translation. These applications aim at producing new data with desirable style attributes (e.g., sentiment, tense, or hair color) while preserving the other core information (e.g., topic or identity) [Li et al., 2019, 2022, Xie et al., 2023, Isola et al., 2017, Zhu et al., 2017]. Consequently, the central challenge in counterfactual generation is to learn the underlying disentangled representations.

To achieve this goal, prior work leverages either paired data that only differ in style components [Rao and Tetreault, 2018, Shang et al., 2019, Xu et al., 2019b, Wang et al., 2019b], or utilizes style labeling information [John et al., 2019, He et al., 2020, Dathathri et al., 2020, Yang and Klein, 2021, Liu et al., 2022]. However, collecting paired data or labels can be labour-intensive and even infeasible in many real-world applications [Chou et al., 2022, Calderon et al., 2022, Xie et al., 2023]. This has prompted recent work [Kong et al., 2022, Xie et al., 2023] to delve into unsupervised identification of latent variables by tapping into multiple domains. To attain identifiability guarantees, a prevalent assumption made in these works [Kong et al., 2022, Xie et al., 2023] is that the content and the style

---

*Equal Contribution. Work was done when Hanqi Yan was a visiting student at MBZUAI. Correspondence to: Yulan He (yulan.he@kcl.ac.uk) and Kun Zhang (kunz1@cmu.edu).

latent variables are independent of each other. However, this assumption is often violated in practical applications. First, the dependence between content and style variables can be highly pronounced. For example, to express a positive sentiment, words such as "*tasty*" and "*flavor*" are typically used in conjunction with food-related content. In contrast, words like "*thrilling*" are more commonly used with movie-related content [Li et al., 2019, 2022]. Moreover, the dependence between content and style often varies across different distributions. For example, a particular cuisine may be highly favored locally but not well received internationally. This varying dependence between content and style variables poses a significant challenge in obtaining the identifiability guarantee. To the best of our knowledge, this issue has not been addressed in previous studies.

In this paper, we address the identification problem of the latent-variable model that takes into account the varying dependence between content and style (see Fig 1). To this end, we adopt a natural notion of influence sparsity inherent to a range of unstructured data, including natural languages, for which the influences from the content and the style differ in their scopes. Specifically, the influence of the style variable on the text is typically sparser compared to that of the content variable, as it is often localized to a smaller fraction of the words [Li et al., 2018] and plays a secondary role in word selection. For instance, the tense of a sentence is typically reflected in only its verbs which are affected by the sentence content information. Our contributions can be summarised as: 1) We show identification guarantees for both the content and the style components, even when their interdependence varies. This approach removes the necessity for a large number of domains with specific variance properties [Kong et al., 2022, Xie et al., 2023]. 2) Guided by our theoretical findings, we design a doMain AdapTive counTerfactual gEneration model (**MATTE**). It does not require paired data or style annotations but allows style intervention, even across substantially different domains. 3) We validate our theoretical discoveries by demonstrating state-of-the-art performance on the unsupervised style transfer task, which demands representation disentanglement, an integral aspect of counterfactual generation.

## 2 Related work

**Label-free Style Transfer on variation autoencoders (VAEs).** To perform style transfer, existing methods that use parallel or non-parallel labelled data often rely on style annotations to refine the attribute-bearing representations, although some argue that disentanglement is not necessary for style edit [Sudhakar et al., 2019, Wang et al., 2019a, Dai et al., 2019]. In practice, disentangled methods typically employ adversarial objectives to ensure the content encoder remains independent of style information or leverage style discriminators to refine the derived style variables [Hu et al., 2017, Shen et al., 2017, Li et al., 2018, Keskar et al., 2019, John et al., 2019, Sudhakar et al., 2019, Dathathri et al., 2020, Hu et al., 2017, Dathathri et al., 2020, Yang and Klein, 2021, Liu et al., 2022]. Several studies have tackled this task without style labels. Riley et al. [2020] emphasized the implicit style connection between adjacent sentences and used T5 [Raffel et al., 2020] to extract the style vector for conditional generation. CPVAE Xu et al. [2020] split the latent variable into content and style variables and mapped them to a $k$-dimensional simplex for $k$-way sentiment modeling. Our work aligns more closely with CPVAE and follows VAE-based label-free disentangled learning from data-generation perspective Higgins et al. [2017], Kumar et al. [2018], Mathieu et al. [2019], Xu et al. [2020], Mita et al. [2021], Fan et al. [2023].

**Latent-variable identification.** Representation learning serves as the cornerstone for generative models, where the goal is to create representations that effectively capture underlying factors in the disentangled data-generation process. In the realm of linear generative functions, Independent component analysis (ICA) [Comon, 1994, Bell and Sejnowski, 1995] is a classical approach known for its identifiability. However, when dealing with nonlinear functions, ICA is proven unidentifiable without the inclusion of additional assumptions [Hyvärinen and Pajunen, 1999]. To tackle this problem, recent work incorporated supplementary information [Hyvarinen et al., 2019, Sorrenson et al., 2020, Hälvä and Hyvarinen, 2020, Khemakhem et al., 2020, Kong et al., 2022], e.g., class/domain labels. However, these approaches require a number of domains/classes that are twice the number of latent components, which can be unfeasible when dealing with high-dimensional representations. Another line of work [Zimmermann et al., 2022, von Kügelgen et al., 2021, Locatello et al., 2020, Gresele et al., 2019, Kong et al., 2023a,b] leverages paired data (e.g., two rotated versions of the same image) to identify the shared latent factor within the pair. The third line of work [Lachapelle et al., 2022, Yang et al., 2022, Zheng et al., 2022] makes sparsity assumptions on the nonlinear generating function. Although their sparsity assumption alleviates some of the explicit requirements in the previous two types of work, it may not hold for complex data distributions. For instance, in the

case of generating text, each topic-related latent factor may influence a large number of components. Instead, we adopt a *relative* sparsity assumption, where we only require the influence of one subspace to be sparser than the other. Unlike prior work [Zheng et al., 2022], each latent variable is allowed to influence a non-sparse set of components, and the influence can overlap arbitrarily within each subspace. Importantly, we necessitate neither many domains/classes nor paired data as prior work mentioned above.

## 3 Disentangled Representation for Counterfactual Generation

In this section, we discuss the connection between counterfactual generation and the identification of the data-generating process shown in Fig 1.

**Disentangled latent representation.** The data-generating process in Figure 1 can be expressed in Equation 1:

$$\mathbf{c} \sim p(\mathbf{c}|\mathbf{u}), \ \mathbf{s} \sim p(\mathbf{s}|\mathbf{c}, \mathbf{u}), \ \mathbf{x} = g(\mathbf{c}, \mathbf{s}), \quad (1)$$

where the data (e.g., text) $\mathbf{x} \in \mathcal{X}$ are generated by latent variables $\mathbf{z} := [\mathbf{c}, \mathbf{s}] \in \mathcal{Z} \subseteq \mathbb{R}^{d_z}$ through a smooth and invertible generating function $g(\cdot) : \mathcal{Z} \to \mathcal{X}$. The latent space comprises two subspaces: the content variable $\mathbf{c} \in \mathcal{C} \subseteq \mathbb{R}^{d_c}$ and the style variable $\mathbf{s} \in \mathcal{S} \subseteq \mathbb{R}^{d_s}$. We define $\mathbf{c}$ as the description of the main topic, e.g., "We ordered the steak recommended by the waitress,", and $\mathbf{s}$ comprises supplementary details connected to the primary topic, e.g., the sentiment towards the dish, as exemplified in "it was *delicious*!". Consequently, the counterfactual generation task here is to preserve the content information $\mathbf{c}$ while altering the stylistic aspects represented by $\mathbf{s}$.

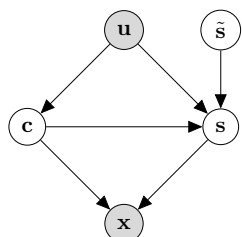

Figure 1: **The data generation process**: The grey shading indicates the variable is observed. The observed variable (i.e., text) $\mathbf{x}$ is generated from content $\mathbf{c}$ and style $\mathbf{s}$. Both content $\mathbf{c}$ and style $\mathbf{s}$ are influenced by the domain variable $\mathbf{u}$ and the content also influences the style. $\tilde{\mathbf{s}}$ is the exogenous variable of $\mathbf{s}$, representing the independent information of $\mathbf{s}$.

**Content-style dependence.** In many real-world problems, content $\mathbf{c}$ can significantly impact style $\mathbf{s}$. For instance, when it comes to the positive descriptions of food (content), words like "*delicious*" are more prevalent than terms like "*effective*". Intuitively, the content $\mathbf{c}$ acts to constrain the range of vocabulary choices for style $\mathbf{s}$. As a result, the counterfactually generated data should preserve the inherent relationship between $\mathbf{c}$ and $\mathbf{s}$. We directly model this dependence as:

$$\mathbf{s} := g_s(\tilde{\mathbf{s}}; \mathbf{c}, \mathbf{u}), \quad (2)$$

where $g_s$ characterizes the influence from $\mathbf{c}$ to $\mathbf{s}$ and the exogenous variable $\tilde{\mathbf{s}}$ accounts for the inherent randomness of $\mathbf{s}$. In the running example, $\tilde{\mathbf{s}}$ can be interpreted as the randomness involved in choosing a word from the vocabulary defined by content $\mathbf{c}$, encompassing words similar to "*delicious*" such as "*tasty*","*yummy*". In contrast, prior work [Kong et al., 2022, Xie et al., 2023] assumes the independence between $\mathbf{c}$ and $\mathbf{s}$ thus neglecting this dependence.

**Challenges from multiple domains.** As we outlined in Section 1, the ability to handle domain shift is crucial for unsupervised counterfactual generation. Domain embedding $\mathbf{u}$ represents a specific domain and the domain distribution shift influences both the marginal distribution of content $p(\mathbf{c}|\mathbf{u})$ and the dependence of content on style $p(\mathbf{s}|\mathbf{c}, \mathbf{u})$. The change in content distribution $p(\mathbf{c}|\mathbf{u})$ across different domains $\mathbf{u}$ reflects the variability in the subjects of sentences across these domains. For example, it can manifest as a change from discussing food in restaurant reviews to actors in movie reviews. The change in content-style dependence $p(\mathbf{s}|\mathbf{c}, \mathbf{u})$ signifies that identical sentence subjects (i.e., content) could be associated with disparate stylistic descriptions in different domains. For instance, the same political question could provoke significantly different sentiments among various demographic groups. Such considerations are absent in prior work [Kong et al., 2022]. Here, we learn a shared model $(\hat{p}(\mathbf{c}|\cdot), g_s(\tilde{\mathbf{s}}, \mathbf{c}, \cdot))$ and domain-specific embeddings $\mathbf{u}$. This approach enables effective knowledge transfer across domains and manages distribution shifts efficiently. For a target domain $\tau$, which may have limited available data, we can learn $\mathbf{u}^\tau$ using a small amount of unlabeled data $\mathbf{x}^\tau$ while preserving the multi-domain information in the shared model.

In light of the above discussion, the essence of counterfactual generation now revolves around the task of discerning the disentangled representation $(\mathbf{c}, \tilde{\mathbf{s}})$ within the data-generating process (Fig 1) across various domains with unlabeled data $(\mathbf{x}, \mathbf{u})$: if we can successfully identify $(\mathbf{c}, \tilde{\mathbf{s}})$, we could perform counterfactual reasoning by manipulating $\tilde{\mathbf{s}}$ while preserving both the content information and the content-style dependence.

# 4 Identifiability of the Latent Variables

In this section, we introduce the identification theory for the content $\mathbf{c}$ and the style $\mathbf{s}$ sequentially and then discuss their implications for methodological development.

We introduce notations and definitions that we use throughout this work. When working with matrices, we adopt the following indexing notations: for a matrix $\mathbf{M}$, the $i$-th row is denoted as $\mathbf{M}_{i,:}$, the $j$-th column is denoted as $\mathbf{M}_{:,j}$, and the $(i,j)$ entry is denoted as $[\mathbf{M}]_{i,j}$. We can also use this notation to refer to specific sets of indices within a matrix. For an index set $\mathcal{I} \subseteq \{1,\ldots,m\} \times \{1,\ldots,n\}$, we use $\mathcal{I}_{i,:}$ to denote the set of column indices whose row coordinate is $i$, i.e., $\mathcal{I}_{i,:} := \{j : (i,j) \in \mathcal{I}\}$, and analogously $\mathcal{I}_{:,j}$ to denote the set of row indices whose column coordinate is $j$.

In addition, we define a subspace of $\mathbb{R}^n$ using an index set $\mathcal{S}$: $\mathbb{R}_{\mathcal{S}}^n = \{\mathbf{z} \in \mathbb{R}^n | \forall i \notin \mathcal{S}, z_i = 0\}$, i.e., it consists of all vectors in $\mathbb{R}^n$ whose entries are zero for all indices not in $\mathcal{S}$. Finally, we can define the support of a matrix-valued function $\mathbf{M}(\mathbf{x}) : \mathcal{X} \to \mathbb{R}^{m \times n}$ as the set of indices whose corresponding entries are nonzero for some input value $\mathbf{x}$, i.e., $\text{Supp}(\mathbf{M}) := \{(i,j) : \exists \mathbf{x} \in \mathcal{X}, \text{s.t.}, [\mathbf{M}(\mathbf{x})]_{i,j} \neq 0\}$.

## 4.1 Influence Sparsity for Content Identification

We show that the subspace $\mathbf{c}$ can be identified. That is, we can estimate a generative model $(p_{\hat{c}}, p_{\hat{s}|\hat{c}}, \hat{g})$ [2] following the data-generating process in Equation 1 and the estimated variable $\hat{\mathbf{c}}$ can capture all the information of $\mathbf{c}$ without interference from $\mathbf{s}$. In the following, we denote the Jacobian matrices $\mathbf{J}_g(\mathbf{z})$'s and $\mathbf{J}_{\hat{g}}(\hat{\mathbf{z}})$'s supports as $\mathcal{G}$ and $\hat{\mathcal{G}}$ respectively. Further, we denote as $\mathcal{T}$ a set of matrices with the same support as that of the matrix-valued function $\mathbf{J}_g^{-1}(\mathbf{c})\mathbf{J}_{\hat{g}}(\hat{\mathbf{c}})$.

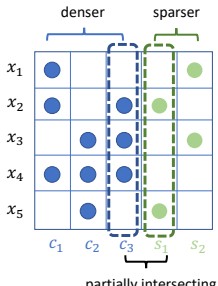

Figure 2: **Sparsity in $\mathbf{J}_g$.**

**Assumption 1** (Content identification).

    *i. $g$ is smooth and invertible and its inverse $g^{-1}$ is also smooth.*

    *ii. For all $i \in \{1,\ldots,d_x\}$, there exist $\{\mathbf{z}^{(\ell)}\}_{\ell=1}^{|\mathcal{G}_{i,:}|}$ and $\mathbf{T} \in \mathcal{T}$, such that $span(\{\mathbf{J}_g(\mathbf{z}^{(\ell)})_{i,:}\}_{\ell=1}^{|\mathcal{G}_{i,:}|}) = \mathbb{R}_{\mathcal{G}_{i,:}}^{d_z}$ and $[\mathbf{J}_g(\mathbf{z}^{(\ell)})\mathbf{T}]_{i,:} \in \mathbb{R}_{\hat{\mathcal{G}}_{i,:}}^{d_z}$.*

    *iii. For every pair $(c_{j_c}, s_{j_s})$ with $j_c \in [d_c]$ and $j_s \in \{d_c + 1, \ldots, d_z\}$, the influence of $s_{j_s}$ is sparser than that of $c_{j_c}$, i.e., $\|\mathcal{G}_{:,j_c}\|_0 > \|\mathcal{G}_{:,j_s}\|_0$.*

**Theorem 1.** *We assume the data-generating process in Equation 1 with Assumption 1. If for given dimensions $(d_c, d_s)$, a generative model $(p_{\hat{c}}, p_{\hat{s}|\hat{c}}, \hat{g})$ follows the same generating process and achieves the following objective:*

$$\underset{p_{\hat{c}}, p_{\hat{s}}, \hat{g}}{\arg\min} \sum_{j_{\hat{s}} \in \{d_c+1,\ldots,d_z\}} \left\|\hat{\mathcal{G}}_{:,j_{\hat{s}}}\right\|_0 \quad \text{subject to: } p_{\hat{\mathbf{x}}}(\mathbf{x}) = p_{\mathbf{x}}(\mathbf{x}), \ \forall \mathbf{x} \in \mathcal{X}, \tag{3}$$

*then the estimated variable $\hat{\mathbf{c}}$ is an one-to-one mapping of the true variable $\mathbf{c}$. That is, there exists an invertible function $h_c(\cdot)$ such that $\hat{\mathbf{c}} = h_c(\mathbf{c})$.*

A proof can be found in Appendix A.5.

**Interpretation.** Theorem 1 states that by matching the marginal distribution $p_{\mathbf{x}}(\mathbf{x})$ under a sparsity constraint of $\hat{\mathbf{s}}$ subspace, we can successfully eliminate the influence of $\mathbf{s}$ from the estimated $\hat{\mathbf{c}}$. This warrants that the content information can be fully retained without being entangled with the style information for a successful counterfactual generation. We can further identify $\tilde{\mathbf{s}}$ from that of $\mathbf{s}$ when the dependence $g_s$ function is invertible in its argument $\tilde{\mathbf{s}}$ [Kong et al., 2022].

**Discussion on assumptions.** Assumption 1-i. ensures that the information of all latent variables $[\mathbf{c}, \mathbf{s}]$ is preserved in the observed variables $\mathbf{x}$, which is a necessary condition for latent-variable identification [Hyvarinen et al., 2019, Kong et al., 2022]. Assumption 1-ii. ensures that the influence from the latent variable $\mathbf{z}$ varies sufficiently over its domain. This excludes degenerate cases where the Jacobian matrix is partially constant, and thus, its support fails to faithfully capture the influence between latent variables and the observed variables. Assumption 1-iii. encodes the observation that the $\mathbf{s}$ subspace exerts a relatively sparser influence on the observed data than the subspace $\mathbf{c}$ (Fig 2.).

---

[2]As our theory is not contingent on the availability of multiple domains, we drop the domain index $\mathbf{u}$ in our notations in this section for ease of exposition.

This is reasonable for language, where the main event largely predominant the sentence and the stylistic variable play complementary and local information for particular attributes, e.g., tense, sentiment, and formality [Xu et al., 2019a, Wang et al., 2021, Ross et al., 2021]. For the language data, $\mathbf{x}$ corresponds to a piece of text (e.g., a sentence) with its dimension $d_{\mathbf{x}}$ equal to the number of words multiplied by the word embedding dimension, i.e., multiple dimensions of $\mathbf{x}$ correspond to a single word. Therefore, even if a word is simultaneously influenced by both $\mathbf{c}$ and $\mathbf{s}$, the influence from the content $\mathbf{c}$ tends to be denser on this word's embedding dimension, as content usually takes precedence in word selection over style.

**Contrast with prior work.** Zheng et al. [2022] impose sparse influence constraints on the generating function $g$ in an absolute sense – each latent component should have a very sparse influence on the observed data. In contrast, Theorem 1 only calls for relative sparsity between two subspaces where each latent component's influence may not be sparse and unique as in Zheng et al. [2022]. We believe this is reasonable for many real-world applications like languages. Kong et al. [2022] assume the independence between the two subspaces and identify the content subspace by resorting to its invariance and sufficient variability of the style subspace over multiple domains. However, as discussed in Section 3, the invariance of the content subspace is often violated, and so is the independence assumption. In contrast, we permit the content subspace to vary over domains and allow for the dependence between the two subspaces.

### 4.2 Partially Intersecting Influences for Style Identification

In this section, we show the identifiability for the style subspace $\mathbf{s}$, under one additional condition: the influences from the two subspaces $\mathbf{c}$ and $\mathbf{s}$ do not fully overlap.

**Assumption 2** (Partially intersecting influence supports). *For every pair $(c_{j_c}, s_{j_s})$, the supports of their influences on $\mathbf{x}$ do not fully intersect, i.e., $\|\mathcal{G}_{:,j_c} \cap \mathcal{G}_{:,j_s}\|_0 < \min\{\|\mathcal{G}_{:,j_c}\|_0, \|\mathcal{G}_{:,j_s}\|_0\}$.*

**Theorem 2.** *We follow the data-generating process Equation 1 and Assumption 1 and Assumption 2. We optimize the objective function in Equation 3 together with*

$$\min \sum_{(j_{\hat{c}}, j_{\hat{s}}) \in \{1,\dots,d_c\} \times \{d_c+1,\dots,d_z\}} \left\| \hat{\mathcal{G}}_{:,j_{\hat{c}}} \cap \hat{\mathcal{G}}_{:,j_{\hat{s}}} \right\|_0. \tag{4}$$

*The estimated style variable $\hat{\mathbf{s}}$ is a one-to-one mapping to the true variable $\mathbf{s}$. That is, there exists an invertible mapping $h_s(\cdot)$ between $\mathbf{s}$ and $\hat{\mathbf{s}}$, i.e., $\hat{\mathbf{s}} = h_s(\mathbf{s})$.*

The proof can be found in Appendix A.6.

**Interpretation.** Theorem 2 states that we can recover the style subspace $\mathbf{s}$ if the influences from the two subspaces do not interfere with each other (Fig 2). This condition endows the subspaces distinguishing footprints and thus forbids the content information in $\mathbf{c}$ from contaminating the estimated style variable $\hat{\mathbf{s}}$. The identification of $\mathbf{s}$ is crucial to counterfactual generation tasks: if the estimated style variable $\hat{\mathbf{s}}$ does capture all the true style variable $\mathbf{s}$, intervening on $\hat{\mathbf{s}}$ cannot fully alter the original style that is intended to be changed.

**Discussion on assumptions.** Assumption 2 prescribes that each content component $c_{j_c}$ and each style component $s_{j_s}$ do not fully contain each other's influence support. Together with Theorem 1, this assumption is essential to the identification of $\mathbf{s}$, without which $\hat{s}_{j_s}$ may absorb the influence from $c_{j_c}$. Assumption 2 does not demand the supports of the entire subspaces $\mathbf{c}$ and $\mathbf{s}$ to be partially intersecting or even disjoint, and the latter directly implies Assumption 2. This assumption is plausible for many real-world data distributions, especially for unstructured data like languages and images – certain dimensions in the pixels and word embeddings may reflect the information of either the content or the style.

**Contrast with prior work.** Kong et al. [2022] obtains the identifiability of the style subspace $\mathbf{s}$ by exploiting the access to multiple domains over which the marginal distribution of $\mathbf{s}$ (i.e., $p(\mathbf{s}|\mathbf{u})$) varies substantially over domains $\mathbf{u}$. This hinges on the independence between the two subspaces and is not applicable when the marginal distribution of $\mathbf{s}$ only varies over the content $\mathbf{c}$, i.e., $p(\mathbf{s}|\mathbf{c}, \mathbf{u})$.

## 5 A Framework for Unsupervised Counterfactual Generation

In this section, we translate the theoretical insights outlined in Section 4 into an unsupervised counterfactual generation framework. Guided by the theory, we can approximate the underlying data-generating process depicted in Fig 1 and recover the disentangled latent components.

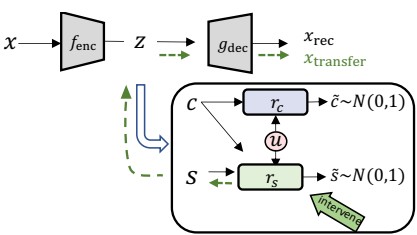

Figure 3: **Our VAE-based framework – MATTE**. During training, the input $\mathbf{x}$ is fed to the encoder $f_{\text{enc}}$ to derive the latent variable $\mathbf{z} = [\mathbf{c}, \mathbf{s}]$, which is then passed to the decoder $g_{\text{dec}}$ for reconstruction. Flow modules, denoted as $r_c$ and $r_s$, are implemented to model the causal influences on $\mathbf{c}$ and $\mathbf{s}$ respectively, which yields the creation of exogenous variables $\tilde{\mathbf{c}}$ and $\tilde{\mathbf{s}}$. To generate transferred data $\mathbf{x}_{\text{transfer}}$, we intervene on the style exogenous variable $\tilde{\mathbf{s}}$ while keeping the original content variable $\mathbf{c}$ unchanged (indicated by the green arrows).

In the following, we will describe each module in our VAE-based estimation framework (Fig 3), the learning objective, and the procedure for counterfactual generation.

### 5.1 VAE-based Estimation Framework

Given input sentences $\mathbf{x}$ from various domains, we use the VAE encoder $f_{\text{enc}}$ to parameterize the posterior distribution $q_{f_{\text{enc}}}(\mathbf{z}|\mathbf{x})$ and sample $\mathbf{z} \sim q_{f_{\text{enc}}}(\mathbf{z}|\mathbf{x})$. [3] The posterior sample $\mathbf{z}$ is then fed into the VAE decoder $g_{\text{dec}}$ for reconstruction $\mathbf{x}_{\text{rec}} = g_{\text{dec}}(\mathbf{z})$, as in conventional VAE training.

We split $\mathbf{z}$ into two components: $\mathbf{c}$ and $\mathbf{s}$. As shown in Fig 1, both $\mathbf{c}$ and $\mathbf{s}$ encompass information of a particular domain $\mathbf{u}$, and $\mathbf{s}$ is also influenced by $\mathbf{c}$. We parameterize such influences using flow-based models [Dolatabadi et al., 2020, Durkan et al., 2019] $r_c$ and $r_s$, respectively:

$$\tilde{\mathbf{c}} = r_c(\mathbf{c}; \mathbf{u}), \ \tilde{\mathbf{s}} = r_s(\mathbf{s}; \mathbf{u}, \mathbf{c}), \tag{5}$$

where $\tilde{\mathbf{c}}$ and $\tilde{\mathbf{s}}$ are exogenous variables that are independent of each other, and $\mathbf{u}$ and $(\mathbf{u}, \mathbf{c})$ act as contextual information for the flow models $r_c(\cdot; \mathbf{u})$ and $r_s(\cdot; \mathbf{u}, \mathbf{c})$. This design promotes parameter sharing across domains, as we only need to learn a domain embedding $\mathbf{u}$ (c.f., a separate flow model per domain). As part of the evidence-lower-bound (ELBO) objective in VAE, we regularize the distributions of $\tilde{\mathbf{z}} = [\tilde{\mathbf{c}}, \tilde{\mathbf{s}}]$ to align with the prior $p(\tilde{\mathbf{z}})$ using Kullback–Leibler (KL) divergence. Consequently, the VAE learning objective can be expressed as:

$$\mathcal{L}_{\text{VAE}} := -\log p_{f_{\text{enc}}, g_{\text{dec}}}(\mathbf{x}_{\text{rec}}) + \text{KL}(q_{f_{\text{enc}}, r_c, r_s}(\tilde{\mathbf{z}}|\mathbf{x})|p(\tilde{\mathbf{z}})), \tag{6}$$

where the prior $p(\tilde{\mathbf{z}})$ is set to a standard Gaussian distribution, $\mathcal{N}(\mathbf{0}, \mathbf{I})$, consistent with typical VAE implementations.

### 5.2 Sparsity Regularization for Identification Guarantees

Guided by the insights from Theorem 1 and Theorem 2, the sparsity constraint on the influence of $\mathbf{s}$ (i.e., Equation 3) and the partially intersecting influence constraint (i.e., Equation 4) are crucial to faithfully recover and disentangle the latent representations $\mathbf{c}$ and $\mathbf{s}$.

**Sparsity of the style influence.** To implement Equation 3, we compute the Jacobian matrix $\mathbf{J}_{g_{\text{dec}}}(\mathbf{z})$ for the decoder function on-the-fly and apply $\ell_1$ regularization to the columns corresponding to the style variable $[\mathbf{J}_{g_{\text{dec}}}(\mathbf{z})]_{:, d_c+1:d_z}$ to control its sparsity. That is, $\mathcal{L}_{\text{sparsity}} = \|[\mathbf{J}_{g_{\text{dec}}}(\mathbf{z})]_{:, d_c+1:d_z}\|_1$.

**Partially intersecting influences.** To encourage sparsity in the intersection of influence between $\mathbf{c}$ and $\mathbf{s}$ (as defined in Equation 4), we select $K$ output dimensions $\mathcal{I}_s$ of $\mathbf{J}_{g_{\text{dec}}}(\mathbf{z})$ that capture the most substantial influence from $\mathbf{s}$ and another set of $K$ output dimensions $\mathcal{I}_c$ that receive the least influence from $\mathbf{c}$. Subsequently, we apply $\ell_1$ regularization to the influence from $\mathbf{c}$ on the output dimensions at the intersection $\mathcal{I}_s \cap \mathcal{I}_c$, i.e., $\mathcal{L}_{\text{partial}} = \|[\mathbf{J}_{g_{\text{dec}}}(\mathbf{z})]_{\mathcal{I}_s \cap \mathcal{I}_c, 1:d_c}\|_1$.

**Content variable masking.** In practice, the content dimensionality $d_c$ is a design choice. When $d_c$ is set excessively large, the sparsity regularization term $\mathcal{L}_{\text{sparsity}}$ may cause the style variable $\mathbf{s}$ to lose its influence, squeezing the information of $\mathbf{s}$ into the content variable $\mathbf{c}$. To handle this issue, we apply a trainable soft mask that operates on $\mathbf{c}$ to dynamically control its dimensionality.

In sum, the overall training objective is as follows:

$$\mathcal{L} := \mathcal{L}_{\text{VAE}} + \lambda_{\text{sparsity}} \cdot \mathcal{L}_{\text{sparsity}} + \lambda_{\text{partial}} \cdot \mathcal{L}_{\text{partial}} + \lambda_{\text{c-mask}} \cdot \mathcal{L}_{\text{c-mask}}, \tag{7}$$

where $\lambda$'s are weight parameters to balance various loss terms.

---

[3]For the sake of simplicity, in this section, we discuss estimated variables without the $\hat{\cdot}$ notation, as in § 4.

## 5.3 Style Intervention

As discussed in Section 3, the content-style dependence should be preserved when generating counterfactual text to ensure linguistic consistency. This can be achieved by intervening on the exogenous style variable $\tilde{s}$ of the original sample. Specifically, we feed the original sample $\mathbf{x}$ to the encoder $f_{\text{enc}}$ to obtain variables $[\mathbf{c}, \mathbf{s}]$. Subsequently, we pass the style variable $\mathbf{s}$ through the flow models $r_s$ to obtain its exogenous counterpart $\tilde{s}$, i.e., $\tilde{s} = r_s(\mathbf{s}; \mathbf{c}, \mathbf{u})$. To carry out style transfer, we set the original variable $\tilde{s}$ to the desired style value $\tilde{s}_{\text{transfer}}$, which is the average of the exogenous style values of randomly selected samples with the desired style. As the flow model $r_s$ is invertible, we can obtain the transferred style variable $\mathbf{s}_{\text{transfer}} = r_s^{-1}(\tilde{s}_{\text{transfer}}; \mathbf{c}, \mathbf{u})$, which, together with the original content variable $\mathbf{c}$, generates the new sample $\mathbf{x}_{\text{transfer}} = g_{\text{dec}}([\mathbf{c}, \mathbf{s}_{\text{transfer}}])$. This process is illustrated in Fig 3 using green arrows. We demonstrate the importance of preserving the content-style dependence and provide evidence that our approach can indeed fulfill this purpose (Fig 4).

## 6 Experimental Results

We validate our theoretical findings by conducting experiments on multiple-domain sentiment transfer tasks, which require effective disentanglement of factors, a concept at the core of our identifiability theory.

**Datasets and Evaluation Schema.** The proposed method is trained on four-domain datasets (Tab 1), i.e, movie (Imdb) [Diao et al., 2014], restaurant (Yelp) [Li et al., 2018], e-commerce (Amazon) [Li et al., 2018] and news (Yahoo) [Zhang et al., 2015, Li et al., 2019]. [4] From common practice [Yang et al., 2018, Lample et al., 2019], we evaluate the generated sentences in terms of the four automatic metrics: (1) **Accuracy**.

Table 1: Dataset on four domains.

| Domains | Train | Dev | Test |
|---|---|---|---|
| IMDB | 344,175 | 27,530 | 27,530 |
| Yelp | 444,102 | 63,484 | 1000 |
| Amazon | 554,998 | 2,000 | 1,000 |
| Yahoo | 4,000 | 4,000 | 4,000 |

We train a CNN classifier on the original style-labelled dataset, which has over 95.0% accuracy when evaluated on the four separate validation datasets. Subsequently, we employ it to evaluate the transformed sentences, gauging how effectively they convey the intended attributes. (2) **BLEU** [Papineni et al., 2002]. It compares the n-grams in the generated text with those in the original text, measuring how well the original content is retained [5]. (3) **G-score**. It represents the geometric mean of the predicted probability for the ground-truth style category and the BLEU score. Due to its comprehensive nature, it is our primary metric. (4) **Fluency**. It is the perplexity score of GPT-2 [Radford et al., 2019] – lower perplexity values indicate a higher levels of fluency. For **human evaluation**, we invited three evaluators proficient in English to rate the sentiment reverse, semantic preservation, fluency and overall transfer quality using a 5-point Likert scale, where higher scores signify better performance. Furthermore, they were asked to rank the generated sentences produced from different models, with the option to include tied items in their ranking.

## 6.1 Sentiment transfer

**Baselines.** We compare our model with the state-of-the-art text transfer models that do not rely on style labels, along with a supervised model, `B-GST` [Sudhakar et al., 2019], which is based on GPT2 [Radford et al., 2019] and accomplishes style transfer through a combination of deletion, retrieval, and generation. The other VAE-based baselines can be divided into two groups based on their architecture: those with LSTM backbones and those utilizing pretrained language models (PLM). Within the LSTM group, $\beta$-`VAE` [Higgins et al., 2017] encourages disentanglement by progressively increasing the latent code capacity. `JointTrain` [Li et al., 2022] uses the GloVe embedding to initialize $\mathbf{s}$ and learns $\mathbf{c}$ through LSTM. `CPVAE` [Xu et al., 2020] is the state-of-the-art unsupervised style transfer model, which maps the style variable to a $k$-dimensional probability simplex to model different style categories. In the PLM group, we use GPT2 [Radford et al., 2019] as the backbone and introduce an additional variational layer after fine-tuning its embedding layer to generate the latent variable $\mathbf{z}$, referred to as `GPT2-FT`. Also, we consider `Optimus` [Li et al., 2020], which is one of the most widely-used pretrained VAE models, utilizing BERT [Devlin et al., 2019] as the encoder and GPT2 as the decoder.

---

[4]Dataset and detailed experiment configurations can be found in Appendix, A.1.

[5]We also adopt CTC score [Deng et al., 2021], to mitigate potential issues brought by the word-overlap measurements in BLEU, as it considers the matching embeddings. The evaluation results are shown in Table 9.

| | IMDB | | | | Yelp | | | |
|---|---|---|---|---|---|---|---|---|
| Model | Acc(↑) | BLEU(↑) | **G-score**(↑) | PPL(↓) | Acc(↑) | BLEU(↑) | **G-score**(↑) | PPL(↓) |
| B-GST [Sudhakar et al., 2019] | 36.20 ±0.80 | 50.45 ±2.62 | 32.09 ±1.81 | 48.58 ±2.08 | 82.00 ±0.20 | 32.06 ±1.34 | 35.43 ±0.92 | 50.45 ±2.38 |
| LSTM β-VAE [Higgins et al., 2017] | **38.27** ±1.03 | 11.37 ±3.03 | 9.05 ±1.19 | **43.59** ±3.07 | 40.30 ±0.92 | 7.58 ±2.73 | 6.86 ±1.08 | 59.34 ±3.81 |
| LSTM JointTrain [Li et al., 2022] | 24.13 ±0.52 | 23.26 ±2.85 | 12.28 ±1.91 | 70.11 ±2.76 | 14.20 ±0.62 | 31.72 ±1.91 | 12.74 ±0.95 | 84.07 ±2.37 |
| LSTM CPVAE [Xu et al., 2020] | 20.15 ±0.40 | **49.82** ±1.25 | **20.01** ±0.96 | 70.18 ±2.78 | 14.50 ±0.30 | 51.47 ±1.81 | 16.84 ±0.92 | 72.81 ±2.26 |
| PLM GPT2-FT [Radford et al., 2019] | 15.20 ±0.25 | 28.93 ±3.16 | 12.19 ±2.84 | 71.08 ±2.19 | 12.00 ±0.42 | 39.62 ±1.92 | 14.49 ±1.35 | 78.37 ±2.14 |
| PLM Optimus [Li et al., 2020] | 14.07 ±0.20 | **59.04** ±1.68 | 17.47 ±1.21 | 61.90 ±2.61 | 13.60 ±0.30 | **69.82** ±1.92 | **21.24** ±1.83 | **52.56** ±2.01 |
| MATTE | **32.43** ±0.28 | 45.10 ±2.91 | **25.92** ±1.62 | **50.08** ±2.02 | **34.30** ±0.26 | **50.14** ±2.51 | **26.34** ±1.37 | **51.51** ±2.09 |

| | Amazon | | | | Yahoo | | | |
|---|---|---|---|---|---|---|---|---|
| Model | Acc(↑) | BLEU(↑) | **G-score**(↑) | PPL(↓) | Acc(↑) | BLEU(↑) | **G-score**(↑) | PPL(↓) |
| B-GST [Sudhakar et al., 2019] | 60.45 ±0.65 | 56.02 ±2.36 | 47.67 ±1.68 | 49.01 ±3.18 | 84.30 ±0.40 | 40.39 ±2.81 | 38.65 ±1.64 | 58.20 ±2.19 |
| LSTM β-VAE Higgins et al. [2017] | **50.08** ±0.68 | 8.04 ±2.62 | 9.39 ±1.24 | **33.09** ±2.53 | **55.47** ±0.40 | 3.77 ±1.32 | 5.85 ±1.71 | **52.17** ±3.06 |
| LSTM JointTrain Li et al. [2022] | 32.90 ±0.42 | 23.21 ±2.16 | 18.33 ±1.07 | 84.63 ±2.76 | 35.33 ±0.28 | 14.04 ±1.72 | 11.62 ±0.92 | 67.34 ±2.84 |
| LSTM CPVAE Xu et al. [2020] | 32.60 ±0.20 | 41.08 ±1.28 | **30.08** ±1.15 | 77.61 ±3.12 | 43.92 ±0.30 | 25.44 ±1.37 | 20.28 ±0.95 | 76.28 ±2.67 |
| PLM GPT2-FT [Radford et al., 2019] | 30.46 ±0.30 | 40.34 ±2.82 | 26.72 ±1.93 | 79.36 ±2.63 | 17.90 ±0.40 | 44.19 ±1.86 | 15.99 ±1.11 | 70.99 ±2.37 |
| PLM Optimus [Li et al., 2020] | 24.80 ±0.20 | **62.50** ±1.55 | 28.53 ±1.22 | 74.66 ±3.10 | 27.10 ±0.15 | 32.73 ±1.82 | 19.17 ±1.69 | 73.18 ±2.76 |
| MATTE | **34.50** ±0.24 | **52.25** ±1.48 | **35.73** ±1.14 | **63.37** ±2.22 | 38.45 ±0.20 | **42.40** ±1.35 | **29.01** ±2.30 | **56.12** ±2.57 |

Table 2: Comparison with unsupervised methods across four domain datasets with supervised B-GST as an upper bound reference. The top and the second-best results are in bold.

**Quantitative performance.** Among LSTM baselines in Table 2, *β-VAE shows high sentiment transfer accuracy and fluency but poor content preservation.* We observed that many generated sentences follow simple but repetitive patterns, e.g., 2.2% transferred sentences in Yelp containing the phrase "I highly recommend" while only 0.6% original sentences do. They are fluent and correctly sentiment flipped but their semantics are significantly different from the original sentences, indicating a generation degradation problem [6]. *CPVAE achieves an overall better G-Score than all the other baselines across three domains (except for Yelp).* Compared with the other LTSM-based methods, its superiority in content preservation is pronounced. *PLMs models achieve overall better BLEU scores compared with the LSTM group.* Optimus outperforms GPT2-FT, which can be partly explained by the fact that the variational layer in Optimus has been pretrained on 1,990K Wikipedia sentences. Our model is built on top of CPVAE with the proposed causal influence modules and sparsity regularisations. It gains consistent improvements in G-score and fluency across all datasets over all the other unsupervised methods. Compared with the supervised method, despite a relatively large gap in accuracy due to the lack of supervision, our approach achieves comparable BLEU scores. The human evaluation results in Table 3 show that *human annotators favour Optimus in terms of content preservation and fluency, but MATTE ranks the best-performing method with 58.5% support set, compared to 41.00% for Optimus.*

| | Style | Content | Fluency | Best rank(%) |
|---|---|---|---|---|
| CPVAE | 1.30 | 2.78 | 3.20 | 21.50 |
| Optimus | 1.41 | **3.79** | **4.12** | 41.00 |
| MATTE | **1.99** | 2.91 | 3.53 | **58.50** |

Table 3: Human evaluation results from three annotators. The Cohen's Kappa coefficient among every two annotators over the Best-rank is 0.46. Human annotators favour Optimus in terms of content preservation and fluency, but MATTE ranks the best-performing method with more than 58% support set after considering the style reverse success rate.

| | |
|---|---|
| Src1: **This guy is an awful actor.** . ✗ | |
| CPVAE | The guy is very **flavorful**. ✓ |
| Optimus | The guy is an **amazing** actor. ✓ |
| MATTE | This guy is an **amazing** actor! ✓ |
| Src2: **I had it a long time now and I still love it.** ✓ | |
| CPVAE | I had it a long time before and I've **never eaten** it. ✗ |
| Optimus | **It is** a long time now and I **always get this food**. ✓ |
| MATTE | I had it a long time now and I **never played** it. ✗ |
| Src3 **These come in handy with those tender special moments**. ✓ | |
| CPVAE | These come in handy with those **sexy care employees**. ✓ |
| Optimus | These **are fateless in their only safe place**. ✗ |
| MATTE | These come in handy with those **poorly executed characteristics**. ✗ |

Table 4: Generated style-transferred Sentences. ✓ , ✗ represent sentiment polarity.

**Qualitative results.** We randomly selected three sentences from the test sets and analyzed the results generated by the top-performing baselines in Table 4. For Src 1, although all the methods successfully transfer the original sentiment of the sentence, CPVAE generates the word '*flavourful*' for the content *guy*, resulting in an unnatural sentence. *This issue arises because CPVAE fails to identify the domain-specific content-style dependency, i.e., the Src domain is in IMDB, while the transferred sentence incorrectly uses the style word 'flavourful' which is commonly used in Yelp.* While Optimus can generate relatively fluent sentences partly due to its powerful decoder, it hardly

---

[6]The *diversity-n* [Li et al., 2016] also indicates repetitive pattern and the evaluation results are in Table 10.

| | IMDB | | | | Yelp | | | |
|---|---|---|---|---|---|---|---|---|
| Model | Acc(↑) | BLEU(↑) | G-score(↑) | PPL(↓) | Acc(↑) | BLEU(↑) | G-score(↑) | PPL(↓) |
| Backbone | 20.15 | 49.82 | 20.01 | 70.18 | 14.50 | 51.47 | 16.84 | 72.81 |
| Indep [Kong et al., 2022] | **45.00**▲ | 30.88 | 19.89 | 61.85▲ | **61.90**▲ | 25.67 | 21.24▲ | 73.78 |
| CausalDep | 28.71▲ | 39.63 | 21.85▲ | 53.25▲ | 22.10▲ | 48.98 | 25.98▲ | 55.14▲ |
| :w. / $\mathcal{L}_{\text{sparsity}}$ | 21.55 | **56.59**△ | 20.90 | 65.26 | 13.20 | **56.26**△ | 14.59 | 54.10△ |
| :w. / $\mathcal{L}_{\text{partial}}$ | 30.18△ | 51.95 | 25.57△ | 54.66 | 33.70△ | 49.09 | 25.81△ | 52.87△ |
| :w. / $\mathcal{L}_{\text{c-mask}}$ (Full) | **32.43**△ | 45.10 | **25.92**△ | **50.08**△ | 34.30△ | 50.14 | **26.34**△ | **51.51**△ |

Table 5: Ablation results on sentiment transfer on two domains. `CausalDep` incorporates style flow $r_s$ to model dependency of **c** on **s**, while `Indep` assumes the independence between the two variables. ▲ marks the improvements over `Backbone`, while △ over the `CausalDep`.

maintains the original semantics (`Src 2, 3`), indicating a lack of effective disentanglement between **c** and **s**. These failure modes demonstrate the importance of a proper disentanglement of content and style and modelling the causal influence between the two across domains. Benefiting from theoretical insights, our approach manages to reflect the content influence across different domains in `Src 1` and retain the content information in `Src2, 3`.

## 6.2 Ablation Study

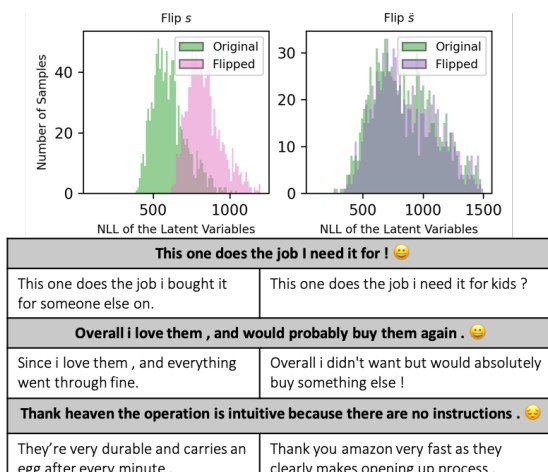

Figure 4: Histograms of negative log-likelihood (NLL) of 1000 Amazon test samples evaluated on the original latent variable and intervened ones. `left`: flips **s**, `right` flips **s̃**. The table shows the corresponding sentences.

Ablation studies in Table 5 are used to verify our theoretical results in § 4 [7]. On top of the backbone, `CPVAE`, we incrementally add each component of our method: (1)`Indep` considers the domain influence on **c** (i.e., the $r_c$ module in Fig 3), while neglecting the independence between **c** and **s**. It experiences a large accuracy boost in conjunction with a significant degradation in BLEU, suggesting poor retention of the content information. (2) `CausalDep` takes into account the dependency between content and style by incorporating the module $r_s$ in Fig 3. This ameliorates the content retention problem and strikes an overall better balance, as reflected by the raised BLEU score and G-score, although causal dependence in `CausalDep` is not sufficient for identification without proper regularization. (3) After introducing the style sparsity regularization $\mathcal{L}_{\text{sparsity}}$ as specified in Theorem 1, we observe a significant increase of BLEU over `CausalDep`, verifying Theorem 1 that the style influence sparsity facilitates *content identification* §4.1. (4) We further introduce $\mathcal{L}_{\text{partial}}$ inspired by Theorem 2, which controls the intersection of content and style influence supports. This improvement in style identification, i.e., the recovery of accuracy over $\mathcal{L}_{\text{sparsity}}$ corroborates Theorem 2. (5) The incorporation of $\mathcal{L}_{\text{c-mask}}$ arrive at our full model, which further improves the style identification, consistent with our motivation in § 5.3. It also exhibits the best G-score across all the datasets, with the most predominant improvement over `CausalDep` on the Yahoo dataset, where the G-score increases from 21.39% to 29.01 %.

**The importance of content-style dependence.** We demonstrate the importance of content-style dependence by visualizing the changes in negative log-likelihood (NLL) induced by different ways of style intervention, namely flipping **s̃** as in our method and flipping **s** which breach the content-style dependence. If the NLL increases after the style transfer, it indicates that the new variables are located in a lower density region [Zheng et al., 2022, Xu et al., 2020]. Fig 4 shows the histograms of NLLs for all the Amazon test samples, both before and after a style transfer. We can see that the NLL distribution changes negligibly when we flip **s̃**, in contrast with the significant change caused by flipping **s**. This implies that flipping **s̃** enables better preservation of the joint distribution of the

---
[7]The results on *Amazon* and *Yahoo* are in Appendix, Table 8.

original sentence. The generated sentences resulting from flipping s̃ exhibit a higher level of semantic fidelity to the original sentence, with a clear inverse sentiment.

## 6.3 Comparison with large language model

As widely recognized, large language models (LLMs) have demonstrated an impressive capability in text generation. However, we consider the principles of counterfactual generation to be complementary to the development of LLMs. We aim to leverage our theoretical insights to further enhance the capabilities of LLMs. We provide examples in Table 6 where LLMs struggle with sentiment transfer, primarily due to their tendency to overlook the broader and implicit sentiments while accurately altering invidivual sentiment words. Consequently, it is reasonable to anticipate that LLMs could benefit from the principles of representation learning, as developed in our work.

| |
| --- |
| *Src:* The buttons to extend the arms worked exactly one time before breaking. |
| *ChatGPT-p1:* The buttons to extend the arms **failed to** work **from the beginning, never functioning even once**. |
| *ChatGPT-p2:* The buttons to extend the arms **never** worked, even once, and **remained functional until they broke**. |
| *Our:* The buttons to extend the arms worked exactly **as described**. |
| *Src:* I love that it uses natural ingredients but it was ineffective on my skin. |
| *ChatGPT-p1:* I **dislike** that it uses natural ingredients, but it was **highly effective** on my skin. |
| *ChatGPT-p2:* I **dislike** that it uses natural ingredients, but it was **highly effective** on my skin. |
| *Our:* I like that it uses natural ingredients, and it was **also good**. |
| *Src:* This case is cute however this is the only good thing about it. |
| *ChatGPT-p1:* This case is **not** cute; however, it is the only good thing about it. |
| *ChatGPT-p2:* This case is **not** cute at all; however, it is the only **bad** thing about it. |
| *Ours:* This case is cute and **overall a valuable product**. |

Table 6: A Sentiment transfer example, on which ChatGPT fails to completely reverse the overall sentiment of the sentence, although it successfully negates individual words within text. In contrast, our method achieves the sentiment reversal with minimal changes. *ChatGPT-p1* and *ChatGPT-p2* represent results obtained from two different prompts, i.e., p1: *"Flip the sentiment of the following sentences, but keep the content unchanged as much as possible."*; p2: *"Please invert the sentiment while preserving content as much as possible in the following sentence that originates from the original domain."*.

## 6.4 Visualization of style variable

We further validate our theoretical insights within additional content-style disentanglement scenarios. As tense has a relatively sparse influence on sentences compared to their content, we choose tense (past and present) as another style for illustration. Specifically, we collect 1000 sentences in either past or present tense from the Yelp Dev set and derive their style representations, denoted as s, by feeding these sentences into our well-trained model. The projection results of `CPVAE` and `MATTE` are shown in 5. The distinct separation between the red and blue data points indicates a more discriminative and better disentangled style variabl. However, in the case of `CPVAE`, some blue data points are mixed within the lower portion of the red region.

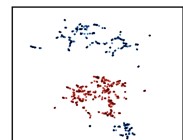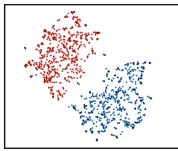

Figure 5: The style variables of sentences in past-tense (blue) and present-tense (red) following a UMAP projection. Left: `CPVAE`; Right: `MATTE`.

## 7 Conclusion and limitations

Prior work [Kong et al., 2022, Xie et al., 2023] have employed multiple domains to achieve unsupervised representation disentanglement. However, the assumed independence between the content and style variables often does not hold in real-world data distributions, particularly in natural languages. To tackle this challenge, we address the identification problem in latent-variable models by leveraging the sparsity structure in the data-generating process. This approach provides identifiability guarantees for both the content and the style variables. We have implemented a controllable text generation method based on these theoretical guarantees. Our method outperforms existing methods on various large-scale style transfer benchmark datasets, thus validating our theory. It is important to note that while our method shows promising empirical results for natural languages, the sparsity assumption (Assumption 1-iii.) may not hold for certain data distributions like images, where the style component could exert dense influences on pixel values. In such cases, we may explore other forms of inherent sparsity in the given distribution, e.g., sparse dependencies between content and style or sparse changes over multiple domains, to achieve identifiability guarantees and develop empirical approaches accordingly.

## Acknowledgements

We thank anonymous reviewers for their constructive feedback. This work was funded by the UK Engineering and Physical Sciences Research Council (grant no. EP/T017112/1, EP/T017112/2, EP/V048597/1, EP/X019063/1). YH is supported by a Turing AI Fellowship funded by the UK Research and Innovation (grant no. EP/V020579/1, EP/V020579/2). The work of LK and YC is supported in part by NSF under the grants CCF-1901199 and DMS-2134080. This project is also partially supported by NSF Grant 2229881, the National Institutes of Health (NIH) under Contract R01HL159805, a grant from Apple Inc., a grant from KDDI Research Inc., and generous gifts from Salesforce Inc., Microsoft Research, and Amazon Research.

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

# A Appendix

## A.1 Implementation Details

In this section, we first introduce the dataset. We then provide the network architecture details of **MATTE**. The hyperparameter selection criteria and the training details are summarized.

### A.1.1 Dataset

We use four domain datasets to train our unsupervised model, i.e., Imdb, Yelp, Amazon and Yahoo, and follow the data split provided by Li et al. [2019]. The datasets can be downloaded via https://github.com/cookielee77/DAST. The dataset details can be found in Table 1. We set the sequence length $L$ as 25, which is the 90 percentile of the sentence length of the training dataset. Therefore, shorter sentences are padded and longer sentences are clipped. The vocabulary size is set to 10000. For the sentiment transfer, we collect 100 positive sentences in dev set based on their sentiment labels to derive $s_{\text{transfer}}$ to flip the sentiment of the negative sentences in the test set, and vice versa. For the tense transfer, we use stanfordnlp tool [8] to identify the tense for the main verb, then collect 100 sentences of present tense in dev set to derive $s_{\text{transfer}}$ to flip the past-tense sentences.

### A.1.2 Model Architecture

We summarize our network architecture below and describe it in detail in Table 7.

**Encoder**: According to Xu et al. [2020], the encoder is fed with a text span $\mathbf{x}[t_1 : t_1 + m]$ extracted from the original sentence $\mathbf{x}$, where $t_1$ is a random word position index, $m$ is set to 10 if $t_1 + m$ is smaller than $L$. $\text{H}_{\text{word}}$ is the word embedding dimension, set to 256. $\text{H}_{\text{lstm}}$ is the hidden states of LSTM, set to 1024. $\text{H}_{\mathbf{z}}$ is the dimension of the latent variable, set to 80. The output of the encoder is the $\mu, \sigma$ and $\mathbf{z}$. All of them are in shape $[\text{BS}, \text{H}_{\mathbf{z}}]$.

**Decoder**: Decoder is fed with the input sentence span and the generated latent variable. The final reconstructed sentence span is one timestamp delay compared to the input span, i.e., $\mathbf{x}_{t_1+1:(t_1+1)+m}$. This is generated by applying beam search to the sequence of output probability over the vocabulary $V$. The $\mathcal{L}_{\text{recon}}$ is to calculate the cross-entropy loss between the output probability and target sequence span.

**Content Flow** $r_c$: We apply Deep Dense Sigmoid Flow (DDSF) [Huang et al., 2018] to derive the content noise term. To incorporate the domain information, we leverage the domain embedding (after MLP) to parameterize the flow model.

**Style Flow** $r_s$: We apply spline flow [Durkan et al., 2019] to derive the noise term. Similarly, we use the conditional flow [9] with extra input. The conditional input is the combination of content variable and domain embedding. Specially, they are concatenated firstly and the result are fed into a MLP with Tanh activation to derive a attention score $\alpha$ The conditional input is actually the doctProdcut.

### A.1.3 Training

**Training details.** The models were implemented in PyTorch 2.0. and Python 3.9. The VAE network is trained for a maximum of 25 epochs and a mini-batch size of 64 is used. We use early stops if the validation reconstruction loss does not decrease for three epochs. For the encoder, we use the Adam optimizer and the learning rate of 0.001. For the decoder, we use SGD with a learning rate of 0.1. For the content and style flow, we use Adam optimizer and the learning rate is 0.001. We set three different random seeds and report the average results.

**Training objective.** The VAE-based model is mainly trained with $\mathcal{L}_{\text{recon}}$ and $\mathcal{L}_{\text{VAE}}$. We use a training trick to better jointly train the other three objectives. The $\mathcal{L}_{\text{sparsity}}$ could cram the information of $\mathbf{s}$ to $\mathbf{c}$, while the $\mathcal{L}_{\text{c-mask}}$ is used to prevent the ill-posed situation where $\mathbf{s}$ have zero influence. Therefore,

---

[8] https://stanfordnlp.github.io/stanfordnlp/

[9] The implementation refers to ConditionedSpline in https://docs.pyro.ai/en/stable/_modules/pyro/distributions/transforms/spline.html

| Module | Description | Output |
|---|---|---|
| **1. Encoder** | Encoder for Input sentence | |
| Input $\mathbf{x}_{t_1:t_1+m}$ | random span of sentence | |
| WordEmb | get word embedding | $\text{BS} \times m \times H_{\text{word}}$ |
| Bi-LSTM | Bi-direction, 2layers | $\text{BS} \times m \times H_{\text{lstm}}$ |
| Average Pooling | sentencet-level Rep. | $\text{BS} \times H_{\text{lstm}}$ |
| MLP | $\mu$ and $\sigma$ | $\text{BS} \times (2 \cdot H_{\mathbf{z}})$ |
| reparameterization | Sampling | $\text{BS} \times H_{\mathbf{z}}$ |
| **2. Domain Embedding** | Embedding Layer | |
| Input u | number of domain $\rightarrow \text{u}_{\text{dim}}$ | $\text{BS} \times \text{u}_{\text{dim}}$ |
| **3. Content Flow $r_c$** | | |
| Input: $\mathbf{c}, \mathbf{u}$ | domain as flow conditional input | |
| MLP | $\mathbf{u} \rightarrow$ conditional context | $\text{BS} \times |\text{H}_{r_c}|$ |
| DDSF | get content noise term $\tilde{\mathbf{c}}$ | $\text{BS} \times \text{c}_{\text{dim}}$ |
| **4. Style Flow $r_s$** | | |
| Input: $\mathbf{c}, \mathbf{u}, \mathbf{s}$ | content and domain as flow context | |
| Concatenate | combine $\mathbf{c}$ and $\mathbf{u}$ | $\text{BS} \times (\text{c}_{\text{dim}} + \text{u}_{\text{dim}})$ |
| MLP | Tanh activation, get attention score $\alpha$ | $\text{BS} \times \text{c}_{\text{dim}}$ |
| Element-wise Multiplication | $\alpha \odot \mathbf{c}$ | $\text{BS} \times \text{c}_{\text{dim}}$ |
| SplineFlow | get style noise term $\tilde{\mathbf{s}}$ | $\text{BS} \times \text{s}_{\text{dim}}$ |
| **5. Decoder $r_s$** | | |
| Input: $\mathbf{z}, \mathbf{x}_{t_1:t_1+m}$ | generate the next token | |
| Bi-LSTM | Bi-direction, 2layers | $\text{BS} \times m \times H_{\text{lstm}}$ |
| MLP | output word probability | $\text{BS} \times m \times \text{V}$ |

Table 7: **MATTE** overall architecture. DDSF is deep dense sigmoid flow, and SplineFlow is neural spline flow. $m$ is the length of randomly extracted text span from input sentence $\mathbf{x}$.

we involve both $\mathcal{L}_{\text{sparsity}}$ and $\mathcal{L}_{\text{c-mask}}$ at the beginning of the training phrase. For $\mathcal{L}_{\text{partial}}$, it is used to sparsify the influence intersection but their separate influences change very frequently in the initial training stages. So we involve it after 3 epochs.

**Computing hardware and running time.** We used a machine with the following CPU specifications: AMD EPYC 7282 CPU. We use NVIDIA GeForce RTX 3090 with 24GB GPU memory. It costs approximately 190ms to run our model on this machine per epoch.

## A.2 Additional Results

This section presents additional results on the hyperparameter sensitivity and the ablation studies on more datasets.

### A.2.1 Hyperparameter Sensitivity

We discuss the effect of the three loss weights $\lambda_{\text{sparsity}}, \lambda_{\text{partial}}$ and $\lambda_{\text{c-mask}}$ in the training objective. We have performed a grid search of $\lambda_{\text{sparsity}} \in [1\text{E-4},1\text{E-3},1\text{E-2}]$, $\lambda_{\text{partial}} \in [3\text{E-5},3\text{E-3},3\text{E-1}]$ and $\lambda_{\text{c-mask}} \in [1\text{E-4},1\text{E-3},1\text{E-2}]$. The best configuration is $[\lambda_{\text{sparisty}}, \lambda_{\text{partial}}, \lambda_{\text{c-mask}}] = [1\text{E-4},3\text{E-3},1\text{E-4}]$. The model performance is relatively sensitive to $\lambda_{\text{sparsity}}$, so we plot the sentiment accuracy and BLEU as a function of $\lambda_{\text{sparsity}}$ in Figure 6.

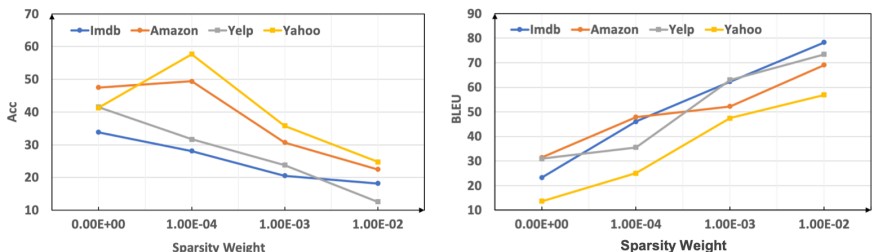

Figure 6: The Sentiment Acc (left) and BLEU (right) with different $\lambda_{\text{sparsity}}$. Among the four datasets, sentiment acc generally decreases as the $\lambda_{\text{sparsity}}$ becomes larger; BLEU increases instead. This observation aligns with our content identifiability theory. We determine the $\lambda_{\text{sparsity}}$ with the best G-score, i.e., 1E-4.

### A.2.2 Ablation Results on Amazon and Yahoo Datasets

We show the ablation study of the Amazon and Yahoo datasets in Table 8. The full model achieves the best G-score and PPL on the two datasets. CausalDep improves the BLEU and PPL. $\lambda_{\text{sparsity}}$ greatly improves the content preservation at the cost of sentiment acc. After incorporating the $\mathcal{L}_{\text{partial}}$ and $\mathcal{L}_{\text{c-mask}}$, the sentiment acc is recovered.

| | Amazon | | | | Yahoo | | | |
|---|---|---|---|---|---|---|---|---|
| Model | Acc(↑) | BLEU(↑) | G-score(↑) | PPL(↓) | Acc(↑) | BLEU(↑) | G-score(↑) | PPL(↓) |
| Backbone | 32.60 | 41.08 | 30.08 | 77.61 | 43.92 | 25.44 | 20.28 | 76.28 |
| Indep [Kong et al., 2022] | **48.80**▲ | 39.50 | 31.76▲ | 77.95 | **51.70** | 23.44 | 21.12▲ | 56.95▲ |
| CausalDep | 33.50▲ | 45.25▲ | 32.54▲ | 66.98▲ | 41.50 | 31.55▲ | 21.39▲ | 64.29▲ |
| :w. / $\mathcal{L}_{\text{sparsity}}$ | 27.10 | **62.73**△ | 34.73△ | **63.37**△ | 27.32 | **48.21**△ | 25.16△ | 60.03△ |
| :w. / $\mathcal{L}_{\text{partial}}$ | 33.10 | 58.54△ | 34.04 | 64.42 | 38.12△ | 41.74 | 28.03△ | 58.04△ |
| :w. / $\mathcal{L}_{\text{c-mask}}$(Full) | 34.50△ | 52.25 | **35.73**△ | 63.37△ | 38.45△ | 42.40△ | **29.01**△ | **56.12**△ |

Table 8: Ablation results on sentiment transfer on two domains. CausalDep incorporates style flow $r_s$ to model dependency of **c** on **s**, while Indep assumes the independence between the two variables. ▲ marks the improvements over Backbone, while △ over the CausalDep.

### A.3 Semantic Preservation Measurement by CTC score

As BLEU has limitations in capturing semantic relatedness beyond literal word-level overlap, we adopt CTC score [Deng et al., 2021] as a complementary evaluation for semantics preservation measurement. For the semantics alignment from $a$ to $b$, CTC considers the matching embeddings, i.e., maximum cosine similarity of all the tokens in $a$ with the tokens in $b$, and vice versa. Then, the final semantic preservation is in *F1*-style definition with one direction result as precision, and the other one as recall. The evaluation results of all the baselines and MATTE are shown in Table 9. The CTC score still favours Optimus and MATTE, with most inferior results on $\beta$-VAE, which are similar trends under the BLEU evaluation schema. Admittedly, the CTC score differences are less discriminative than BLEU – this phoneme is also observed in Liu et al. [2022].

### A.4 Diversity measurements for generated sentences

To further demonstrate the generation degradation issue–generate oversimplified and repetitious sentences, we use diversity-2 [Li et al., 2016], the ratio of distinct two-grams in all the two-grams in the generated sentences to evaluate the transferred sentences. The diversity-2 for original sentences is also included for better comparison. The results in Table 10 show that all the other methods except for $\beta$-VAE generated sentences with similar diversity-2 as the original sentences, but the sentences generated by $\beta$-VAE have much lower diversity than the original ones.

|        | IMDB  | Yelp  | Amazon | Yahoo |
|--------|-------|-------|--------|-------|
| BGST   | **0.468** | 0.458 | **0.472** | **0.458** |
| $\beta$-VAE | 0.436 | 0.433 | 0.433 | 0.413 |
| JointTrain | 0.456 | 0.462 | 0.455 | 0.437 |
| CPVAE | 0.462 | 0.463 | 0.461 | 0.443 |
| GPT2-FT | 0.459 | 0.459 | 0.458 | 0.448 |
| Optimus | **0.465** | **0.468** | 0.465 | 0.446 |
| Matte | **0.465** | **0.464** | **0.466** | **0.452** |

Table 9: CTC score, a complementary evaluation for semantics preservation. $\beta$-VAE displays the least impressive performance, and Optimus and Matte exhibit the overall best results.

| Dataset | IMDB (0.34) | Yelp (0.63) | Amazon (0.64) | Yahoo(0.44) |
|---------|-------------|-------------|---------------|-------------|
| $\beta$-VAE | 0.11 | 0.46 | 0.37 | 0.22 |
| JointTrain | 0.21 | 0.59 | 0.56 | 0.37 |
| CPVAE | 0.32 | 0.59 | 0.57 | 0.45 |
| MATTE | 0.32 | 0.62 | 0.61 | 0.45 |

Table 10: Diversity-2 for the transferred sentences. Diversity for the original sentences is included in the bracket for comparison. $\beta$-VAE has significantly fewer distinct 2-gram than original datasets. This results are consistent with evaluation results on BLEU.

## A.5 Proof for Theorem 1

The original Assumption 1 and Theorem 1 are copied below for reference.

**Assumption 1** (Content identification)**.**

  i. $g$ is smooth and invertible and its inverse $g^{-1}$ is also smooth.

  ii. For all $i \in \{1, \ldots, d_x\}$, there exist $\{\mathbf{z}^{(\ell)}\}_{\ell=1}^{|\mathcal{G}_{i,:}|}$ and $\mathbf{T} \in \mathcal{T}$, such that $span(\{\mathbf{J}_g(\mathbf{z}^{(\ell)})_{i,:}\}_{\ell=1}^{|\mathcal{G}_{i,:}|}) = \mathbb{R}_{\mathcal{G}_{i,:}}^{d_z}$ and $[\mathbf{J}_g(\mathbf{z}^{(\ell)})\mathbf{T}]_{i,:} \in \mathbb{R}_{\hat{\mathcal{G}}_{i,:}}^{d_z}$.

  iii. For every pair $(c_{j_c}, s_{j_s})$ with $j_c \in [d_c]$ and $j_s \in \{d_c + 1, \ldots, d_z\}$, the influence of $s_{j_s}$ is sparser than that of $c_{j_c}$, i.e., $\|\mathcal{G}_{:,j_c}\|_0 > \|\mathcal{G}_{:,j_s}\|_0$.

**Theorem 1.** *We assume the data-generating process in Equation 1 with Assumption 1. If for given dimensions $(d_c, d_s)$, a generative model $(p_{\hat{c}}, p_{\hat{s}|\hat{c}}, \hat{g})$ follows the same generating process and achieves the following objective:*

$$\underset{p_{\hat{c}}, p_{\hat{s}}, \hat{g}}{\arg\min} \sum_{j_{\hat{s}} \in \{d_c+1, \ldots, d_z\}} \left\|\hat{\mathcal{G}}_{:,j_{\hat{s}}}\right\|_0 \quad subject\ to:\ p_{\hat{\mathbf{x}}}(\mathbf{x}) = p_{\mathbf{x}}(\mathbf{x}),\ \forall \mathbf{x} \in \mathcal{X}, \tag{3}$$

*then the estimated variable $\hat{\mathbf{c}}$ is an one-to-one mapping of the true variable $\mathbf{c}$. That is, there exists an invertible function $h_c(\cdot)$ such that $\hat{\mathbf{c}} = h_c(\mathbf{c})$.*

*Proof.* We first define the notation $\mathbf{z} = [\mathbf{c}, \mathbf{s}]$ and the indeterminacy function:

$$h := \hat{g}^{-1} \circ g,$$

which is an invertible function $h : \mathcal{Z} \to \hat{\mathcal{Z}}$ as $g$ is invertible by Assumption 1-i.. According to the chain rule, we have the following relation among the Jacobian matrices:

$$\mathbf{J}_{\hat{g}}(\hat{\mathbf{z}}) = \mathbf{J}_g(\mathbf{z})\mathbf{J}_h^{-1}(\mathbf{z}). \tag{8}$$

We define the support notations as follows:

$$\mathcal{G} := \mathrm{supp}(\mathbf{J}_g(\mathbf{z})),$$
$$\hat{\mathcal{G}} := \mathrm{supp}(\mathbf{J}_{\hat{g}}(\hat{\mathbf{z}})),$$
$$\mathcal{T} := \mathrm{supp}(\mathbf{J}_h^{-1}(\mathbf{z})).$$

In the following, we will show that $(j_c, j_{\hat{s}}) \notin \mathcal{T}$ for any $j_c \in \{1, \ldots, d_c\}$ and $j_{\hat{s}} \in \{d_c + 1, \ldots, d_c + d_s\}$. That is, $[\mathbf{J}_h^{-1}(\mathbf{z})]_{j_c, j_{\hat{s}}} = 0$, for any $j_c \in \{1, \ldots, d_c\}$ and $j_{\hat{s}} \in \{d_c + 1, \ldots, d_c + d_s\}$, which implies that $\mathbf{c}$ is not influenced by $\hat{\mathbf{s}}$.

Because of Assumption 1-ii., for any $i \in \{1, \ldots, d_{v_1} + d_{v_2}\}$, there exists $\{\mathbf{z}^{(\ell)}\}_{\ell=1}^{|\mathcal{G}_{i,:}|}$, such that $\text{span}(\{\mathbf{J}_g(\mathbf{z}^{(\ell)})_{i,:}\}_{\ell=1}^{|\mathcal{G}_{i,:}|}) = \mathbb{R}^{d_z}_{\mathcal{G}_{i,:}}$.

Since $\{\mathbf{J}_g(\mathbf{z}^{(\ell)})_{i,:}\}_{\ell=1}^{|\mathcal{G}_{i,:}|}$ forms a basis of $\mathbb{R}^{d_z}_{\mathcal{G}_{i,:}}$, for any $j_0 \in \mathcal{G}_{i,:}$, we can express canonical basis vector $\mathbf{e}_{j_0} \in \mathbb{R}^{d_z}_{\mathcal{G}_{i,:}}$ as:

$$\mathbf{e}_{j_0} = \sum_{\ell \in \mathcal{G}_{i,:}} \alpha_\ell \cdot \mathbf{J}_g(\mathbf{z}^{(\ell)})_{i,:}, \tag{9}$$

where $\alpha_\ell \in \mathbb{R}$ is a coefficient.

Also, following Assumption 1-ii., there exists a deterministic matrix $\mathbf{T}$ where $\mathbf{T}_{j_1, j_2} \neq 0$ iff $(j_1, j_2) \in \mathcal{T}$ and

$$\mathbf{T}_{j_0,:} = \mathbf{e}_{j_0}^\top \mathbf{T} = \sum_{\ell \in \mathcal{G}_{i,:}} \alpha_\ell \cdot \mathbf{J}_g(\mathbf{z}^{(\ell)})_{i,:} \mathbf{T} \in \mathbb{R}^{d_z}_{\hat{\mathcal{G}}_{i,:}}, \tag{10}$$

where $\in$ is because each element in the summation belongs to $\mathbb{R}^{d_z}_{\hat{\mathcal{G}}_{i,:}}$.

Therefore,

$$\forall j \in \mathcal{G}_{i,:}, \mathbf{T}_{j,:} \in \mathbb{R}^{d_z}_{\hat{\mathcal{G}}_{i,:}}.$$

Equivalently, we have:

$$\forall (i, j) \in \mathcal{G}, \quad \{i\} \times \mathcal{T}_{j,:} \subset \hat{\mathcal{G}}. \tag{11}$$

As both $\mathbf{J}_g$ and $\mathbf{J}_{\hat{g}}$ are invertible, $\mathbf{J}_h(\mathbf{z})$ is an invertible matrix and thus has a non-zero determinant. Expressing $\mathbf{J}_h(\mathbf{z})$ with the Leibniz formulae gives:

$$\det(\mathbf{J}_h(\mathbf{z})) = \sum_{\sigma \in \mathcal{P}_{d_z}} \left( \text{sign}(\sigma) \prod_{j=1}^{d_z} \mathbf{J}_g(\mathbf{z})_{\sigma(j), j} \right) \neq 0, \tag{12}$$

where $\mathcal{P}_{d_z}$ is the set of all $d_z$-permutations.

Equation 12 indicates that there exists $\sigma \in \mathcal{P}_{d_z}$, such that $\prod_{j=1}^{d_z} \mathbf{J}_g(\mathbf{z})_{\sigma(j), j} \neq 0$. Equivalently, we have

$$\forall j \in [d_z], (\sigma(j), j) \in \mathcal{T}. \tag{13}$$

Therefore, for a specific $j_{\hat{s}} \in \{d_c + 1, \ldots, d_z\}$, it follows that $(\sigma(j_{\hat{s}}), j_{\hat{s}}) \in \mathcal{T}$. Further, Equation 11 shows that for any $i_x \in [d_x]$, s.t., $(i_x, \sigma(j_{\hat{s}})) \in \mathcal{G}$, we have $\{i_x\} \times \mathcal{T}_{\sigma(j_{\hat{s}}),:} \subseteq \hat{\mathcal{G}}$. Together, it follows that

$$(i_x, \sigma(j_{\hat{s}})) \in \mathcal{G} \implies (i_x, j_{\hat{s}}) \in \hat{\mathcal{G}}. \tag{14}$$

Equation 14 suggests that the column $\sigma(j_{\hat{s}})$ of the true generating function support $\mathcal{G}$ is included in the column $j_{\hat{s}}$ of the estimated generating function support $\hat{\mathcal{G}}$. Together with Assumption 1-iii., it follows that

$$\sum_{j_{\hat{s}} \in \{d_c+1, \ldots, d_z\}} \left\| \hat{\mathcal{G}}_{:,j_{\hat{s}}} \right\|_0 \geq \sum_{j_s \in \{d_c+1, \ldots, d_z\}} \left\| \mathcal{G}_{:,j_s} \right\|_0, \tag{15}$$

where the permutation $\sigma(\cdot)$ connects the indices of $\mathbf{s}$ and those of $\hat{\mathbf{s}}$. We note that Equation 15 is a lower-bound of the objective Equation 3, which can be attained by a minimizer $\hat{g} = g$.

In the following, we show by contradiction that the support of $\mathbf{J}_h^{-1}(\mathbf{z})$ does not contain $(j_c, j_{\hat{s}})$, for any $j_c \in [d_c]$ and any $j_{\hat{s}} \in \{d_c + 1, \ldots, d_c\}$, i.e., $(j_c, j_{\hat{s}}) \notin \mathcal{T}$.

We suppose that a specific $(j'_c, j'_{\hat{s}}) \in \mathcal{T}$, where $j'_c \in [d_c]$ and any $j'_{\hat{s}} \in \{d_c + 1, \ldots, d_c\}$. We note that the argument for Equation 14 also applies to $(j_1, j_2) \in \mathcal{T}$ for any $j_1, j_2 \in [d_z]$. Thus, we would have

$$(j'_c, j'_{\hat{s}}) \in \mathcal{T} \implies (i_x, j'_{\hat{s}}) \in \hat{\mathcal{G}}, \quad \forall i_x \in \{i \in [d_x] : (i_x, j'_c) \in \mathcal{G}\}. \tag{16}$$

It would follow that

$$\sum_{j_{\hat{s}} \in \{d_c+1,\ldots,d_z\} \setminus \{j'_{\hat{s}}\}} \left\| \hat{\mathcal{G}}_{:,j_{\hat{s}}} \right\|_0 + \left\| \hat{\mathcal{G}}_{:,j'_{\hat{s}}} \right\|_0 \geq \sum_{j_{\hat{s}} \in \{d_c+1,\ldots,d_z\} \setminus \{j'_{\hat{s}}\}} \left\| \mathcal{G}_{:,\sigma(j_{\hat{s}})} \right\|_0 + \left\| \hat{\mathcal{G}}_{:,j'_{\hat{s}}} \right\|_0$$

$$\geq \sum_{j_{\hat{s}} \in \{d_c+1,\ldots,d_z\} \setminus \{j'_{\hat{s}}\}} \left\| \mathcal{G}_{:,\sigma(j_{\hat{s}})} \right\|_0 + \left\| \mathcal{G}_{:,\sigma(j'_{\hat{s}})} \cup \mathcal{G}_{:,j'_c} \right\|_0$$

$$\geq \sum_{j_{\hat{s}} \in \{d_c+1,\ldots,d_z\} \setminus \{j'_{\hat{s}}\}} \left\| \mathcal{G}_{:,\sigma(j_{\hat{s}})} \right\|_0 + \left\| \mathcal{G}_{:,j'_c} \right\|_0$$

$$\underbrace{>}_{(1)} \sum_{j_s \in \{d_c+1,\ldots,d_z\}} \left\| \mathcal{G}_{:,j_s} \right\|_0,$$

$$\tag{17}$$

where the inequality (1) is due to Assumption 1-iii. that the influence of $\mathbf{c}$ on $\mathbf{x}$ is denser than that of $\mathbf{s}$.

However, as discussed above, there exists an optimizer that attains the lower-bound Equation 15. Equation 17 contradicts the minimization objective Equation 3. Therefore, $(j'_c, j'_{\hat{s}}) \notin \mathcal{T}$, for any $j'_c \in [d_c]$ and any $j'_{\hat{s}} \in \{d_c + 1, \ldots, d_c\}$.

As discussed above, this implies that $\mathbf{c}$ is not influenced by $\hat{\mathbf{s}}$. Further, it follows from the invertibility of $h(\cdot)$ that $[\mathbf{J}_h(\mathbf{z})]_{j_{\hat{c}}, j_s} = 0$, for any $j_{\hat{c}} \in \{1, \ldots, d_c\}$ and $j_s \in \{d_c + 1, \ldots, d_c + d_s\}$, which implies that $\hat{\mathbf{c}}$ is not influenced by $\mathbf{s}$. These two conditions and the invertibility of $h(\cdot)$ imply that $\hat{\mathbf{c}}$ and $\mathbf{c}$ form a one-to-one mapping.

$\square$

## A.6 Proof for Theorem 2

The original Assumption iii. and Theorem 2 are copied below for reference.

**Assumption 2** (Partially intersecting influence supports). *For every pair $(c_{j_c}, s_{j_s})$, the supports of their influences on $\mathbf{x}$ do not fully intersect, i.e., $\|\mathcal{G}_{:,j_c} \cap \mathcal{G}_{:,j_s}\|_0 < \min\{\|\mathcal{G}_{:,j_c}\|_0, \|\mathcal{G}_{:,j_s}\|_0\}$.*

**Theorem 2.** *We follow the data-generating process Equation 1 and Assumption 1 and Assumption 2. We optimize the objective function in Equation 3 together with*

$$\min \sum_{(j_{\hat{c}}, j_{\hat{s}}) \in \{1,\ldots,d_c\} \times \{d_c+1,\ldots,d_z\}} \left\| \hat{\mathcal{G}}_{:,j_{\hat{c}}} \cap \hat{\mathcal{G}}_{:,j_{\hat{s}}} \right\|_0. \tag{4}$$

*The estimated style variable $\hat{\mathbf{s}}$ is a one-to-one mapping to the true variable $\mathbf{s}$. That is, there exists an invertible mapping $h_s(\cdot)$ between $\mathbf{s}$ and $\hat{\mathbf{s}}$, i.e., $\hat{\mathbf{s}} = h_s(\mathbf{s})$.*

*Proof.* As shown in Section A.5, there exists a $d_z$-permutation $\sigma(\cdot)$ such that $\forall j \in [d_z], (\sigma(j), j) \in \mathcal{T}$. Also, we have shown in Theorem 1 that $(j_c, j_{\hat{s}}) \notin \mathcal{T}$ for $j_c \in [d_c]$ and $j_{\hat{s}} \in \{d_c + 1, \ldots, d_z\}$, which implies that $\sigma(j_{\hat{s}}) \in \{d_c + 1, \ldots, d_z\}$. Thus, it follows that for any $j_{\hat{c}} \in [d_c], \sigma(j_{\hat{c}}) \in [d_c]$.

In the following, we show by contradiction that $(j_s, j_{\hat{c}}) \notin \mathcal{T}$ for any $j_s \in \{d_c + 1, \ldots, d_z\}$ and $j_{\hat{c}} \in [d_c]$. We suppose that $(j'_s, j'_{\hat{c}}) \in \mathcal{T}$. Analogous to Equation 16, we would have that

$$(j'_s, j'_{\hat{c}}) \in \mathcal{T} \implies (i_x, j'_{\hat{c}}) \in \hat{\mathcal{G}}, \quad \forall i_x \in \{i \in [d_x] : (i_x, j'_s) \in \mathcal{G}\}. \tag{18}$$

It would follow that $\hat{\mathcal{G}}_{:,j'_{\hat{c}}} \supseteq \mathcal{G}_{:,\sigma(j'_{\hat{c}})} \cup \mathcal{G}_{:,j'_s}$. Also, to attain the objective Equation 3 in Theorem 1, we have $j'_{\hat{s}} := \sigma^{-1}(j'_s) \in \{d_c + 1, \ldots, d_z\}$, s.t., $\hat{\mathcal{G}}_{:,j'_{\hat{s}}} = \mathcal{G}_{:,j'_s}$. It would follow that $\hat{\mathcal{G}}_{:,j'_{\hat{c}}} \supseteq \hat{\mathcal{G}}_{:,j'_{\hat{s}}}$.

Further, we would have

$$\left\| \hat{\mathcal{G}}_{:,j'_{\hat{c}}} \cap \hat{\mathcal{G}}_{:,j'_{\hat{s}}} \right\|_0 = \left\| \hat{\mathcal{G}}_{:,j'_{\hat{s}}} \right\|_0 = \left\| \mathcal{G}_{:,j'_s} \right\|_0 \underbrace{>}_{(2)} \left\| \mathcal{G}_{:,\sigma(j'_{\hat{c}})} \cap \mathcal{G}_{:,\sigma(j'_s)} \right\|_0, \tag{19}$$

where (2) is due to Assumption 2.

We note that the lower-bound for Equation 4 is

$$\sum_{(j_{\hat{c}},j_{\hat{s}})\in\{1,\ldots,d_c\}\times\{d_c+1,\ldots,d_z\}}\left\|\hat{\mathcal{G}}_{:,j_{\hat{c}}}\cap\hat{\mathcal{G}}_{:,j_{\hat{s}}}\right\|_0 \geq \sum_{(j_{\hat{c}},j_{\hat{s}})\in\{1,\ldots,d_c\}\times\{d_c+1,\ldots,d_z\}}\left\|\mathcal{G}_{:,\sigma(j_{\hat{c}})}\cap\mathcal{G}_{:,\sigma(j_{\hat{s}})}\right\|_0$$
(20)

$$= \sum_{(j_c,j_s)\in\{1,\ldots,d_c\}\times\{d_c+1,\ldots,d_z\}}\left\|\mathcal{G}_{:,j_c}\cap\mathcal{G}_{:,j_s}\right\|_0,$$
(21)

which can be achieved by $\mathcal{G}=\hat{\mathcal{G}}$. Note that the lower-bounds for both Equation 3 and Equation 4 can be attained simultaneously by $\mathcal{G}=\hat{\mathcal{G}}$. Hence, optimizing the sum of the two objectives does not alter the optimal value of either.

Applying a similar argument as that in Equation 15, we would have that

$$\sum_{(j_{\hat{c}},j_{\hat{s}})\in\{1,\ldots,d_c\}\times\{d_c+1,\ldots,d_z\}}\left\|\hat{\mathcal{G}}_{:,j_{\hat{c}}}\cap\hat{\mathcal{G}}_{:,j_{\hat{s}}}\right\|_0$$

$$= \sum_{(j_{\hat{c}},j_{\hat{s}})\in\{1,\ldots,d_c\}\times\{d_c+1,\ldots,d_z\}\setminus\{(j'_{\hat{c}},j'_{\hat{s}})\}}\left\|\hat{\mathcal{G}}_{:,j_{\hat{c}}}\cap\hat{\mathcal{G}}_{:,j_{\hat{s}}}\right\|_0 + \left\|\hat{\mathcal{G}}_{:,j'_{\hat{c}}}\cap\hat{\mathcal{G}}_{:,j'_{\hat{s}}}\right\|_0$$

$$\geq \sum_{(j_{\hat{c}},j_{\hat{s}})\in\{1,\ldots,d_c\}\times\{d_c+1,\ldots,d_z\}\setminus\{(j'_{\hat{c}},j'_{\hat{s}})\}}\left\|\mathcal{G}_{:,\sigma(j_{\hat{c}})}\cap\mathcal{G}_{:,\sigma(j_{\hat{s}})}\right\|_0 + \left\|\hat{\mathcal{G}}_{:,j'_{\hat{c}}}\cap\hat{\mathcal{G}}_{:,j'_{\hat{s}}}\right\|_0$$
(22)

$$\underset{(3)}{>} \sum_{(j_{\hat{c}},j_{\hat{s}})\in\{1,\ldots,d_c\}\times\{d_c+1,\ldots,d_z\}\setminus\{(j'_{\hat{c}},j'_{\hat{s}})\}}\left\|\mathcal{G}_{:,\sigma(j_{\hat{c}})}\cap\mathcal{G}_{:,\sigma(j_{\hat{s}})}\right\|_0 + \left\|\mathcal{G}_{:,\sigma(j'_{\hat{c}})}\cap\mathcal{G}_{:,\sigma(j'_{\hat{s}})}\right\|_0$$

$$= \sum_{(j_c,j_s)\in\{1,\ldots,d_c\}\times\{d_c+1,\ldots,d_z\}}\left\|\mathcal{G}_{:,j_c}\cap\mathcal{G}_{:,j_s}\right\|_0,$$

where (3) is due to Equation 19. Hence, this was not the minimizer of Equation 4. By contradiction, we have that $(j_s,j_{\hat{c}})\notin\mathcal{T}$ for any $j_s\in\{d_c+1,\ldots,d_z\}$ and $j_{\hat{c}}\in[d_c]$. This implies that $\mathbf{s}$ is not influenced by $\hat{\mathbf{c}}$. Further, it follows from the invertibility of $h(\cdot)$ that $[\mathbf{J}_h(\mathbf{z})]_{j_{\hat{s}},j_c}=0$, for any $j_{\hat{s}}\in\{d_c+1,\ldots,d_z\}$ and $j_c\in[d_c]$, which implies that $\hat{\mathbf{s}}$ is not influenced by $\mathbf{c}$. These two conditions and the invertibility of $h(\cdot)$ imply that $\hat{\mathbf{s}}$ and $\mathbf{s}$ form a one-to-one mapping.

$\square$

