# OpenReview forum: "Counterfactual Generation with Identifiability Guarantees"
_NeurIPS.cc/2023/Conference — NeurIPS 2023 poster_

### Official Review · Reviewer_yXyx · 2023-06-30

**Soundness:** 4 excellent
**Presentation:** 3 good
**Contribution:** 3 good
**Rating:** 8
**Confidence:** 3

**Summary:**

This paper proposes a framework for unsupervised style transfer by disentangling the content and style representations. Unlike previous research, they do not assume independence between the content and style variables in the generation process but rather only a lower influence of the style variable compared to the content variable on the generation process. Based on this assumption, the paper then introduces two theories with corresponding proofs of the identifiability of both the style and content variables under these assumptions. That is, the content information can be preserved without needing style.

Given the theoretical discussion, the paper then proposes a framework based on variational autoencoders (VAE) to address the task of style transfer through style and content disentanglement. The framework is then evaluated empirically on the task by comparing it to other state-of-the-art baselines. In the results, the paper demonstrates the gain from their implementation in most measures considered. The ablation study further illustrates the importance of every component added to their model.


**Strengths:**

- This paper contributes to the style transfer task by eliminating the independence assumption between style and content.

- It supports its claims by providing both theoretical backing and empirical evidence.

- It is clear in most of its parts.


**Weaknesses:**

I do not see any major weaknesses in this paper. However, the authors can address some clarity issues in the questions section.

**Questions:**

- The submission title differs from the title of the paper. Is that on purpose?

- Line 153-156: If I understood correctly, then the described subspace should consist of all vectors whose entries are *not zero* for all indices in S. but in the paper says *zero for all indices in S*

- In Section 5.3, the paper mentions that the s_(transfer) is computed from the average of randomly sampled style values of the *desired style*. What is the desired style here?

- The paper considers the G-score the most important measure because of its comprehensive assessment. It would be nice if the paper clarified what it means to have a comprehensive assessment.

- In section 6.1, It would be nice if the paper clarified the implemented model and the baselines. I learned later in the results section that the model is built on top of CPAVE

- In Table-2 what are src1,2, and 3

**Limitations:**

- The central limiting aspect of this paper is the assumption that style has a lower influence on the generated text compared to the content, which doesn't hold for all text generation tasks. However, this limitation is acknowledged by the authors.

---

> ### Author Rebuttal · Authors · 2023-08-10
>
> Thank you so much for the valuable comments and suggestions. We also hope that our method could provide insights for the following research. We respond below to address your concerns.
>
> ### Title
> Thank you for letting us know this. The title should be “Counterfactual generation with identifiability guarantee”. We will make it consistent.
>
> ### Subspace and zero-vector
>
> You are totally right! Thank you for spotting this typo; we will correct it in our manuscript.
>
> ### Desired style
>
> Great question! For general sentence transfer, the desired style is the target attribute of the sentence, e.g., positive sentiment, formal writing style, female subject, etc. We calculate the $s_\text{transfer}$ over sentences with the target attribute.
>
>
> ### Comprehensive assessment
>
> Thank you for the good question! There is an inherent trade-off between sentiment transfer and semantic preservation because a larger perturbation could lead to a higher sentiment flip success rate but may distort the original semantics to a greater extent. See Table 2 in [1] for an illustration “$\beta$-VAE under 1/2/3 sigma perturbation”. Therefore, G-score was proposed in [2] as a single metric to simultaneously consider both aspects and reflect the overall quality of the transfer, and it is followed by lots of related work [3,4].
>
> ### Implementation clarification
> Thanks for pointing out this issue. You are right -- our model is built on CPVAE. We will improve the baselines section in the revised manuscript to make it clearer.
>
> ### src in Table-2
>
> src-* refers to the original sentences that express one particular sentiment and will be transferred to new sentences with the opposite sentiment by the listed approaches.
>
>
> Please let us know if you have further concerns -- thank you!
>
> ### References:
>
> [1] On variational learning of controllable representations for text without supervision. ICML20.
>
> [2] Unpaired sentiment-to-sentiment translation: A cycled reinforcement learning approach. ACL18.
>
> [3] Reformulating Unsupervised Style Transfer as Paraphrase Generation. EMNLP20
>
> [4] Deep Learning for Text Style Transfer: A Survey. CL22

---

> > ### Comment · Reviewer_yXyx · 2023-08-14
> >
> > Thanks a lot for clarifying my questions. I hope that these answers get integrated into the final version of the paper.

---

> > > ### Author Response · Authors · 2023-08-14
> > >
> > > We will improve our draft according to your suggestions as indicated above—many thanks for the time and effort dedicated to our work!

---

### Official Review · Reviewer_1Pqp · 2023-07-03

**Soundness:** 3 good
**Presentation:** 2 fair
**Contribution:** 3 good
**Rating:** 6
**Confidence:** 3

**Summary:**

In the paper, the authors take varying dependence between content and style into account in the counterfactual generation process. The identification problem is addressed when using a VAE for the task. It is proved that the subspaces of latent variables for contents and style are identifiable. Then based on the theorems and assumptions the authors propose to build a VAE based model with sparsity regularization to solve the problem. The proposed framework is tested on four datasets and gets relatively high scores compared with other unsupervised baselines. Ablation study and case study are also conducted to consolidate the conclusion.

**Strengths:**

1. Detailed mathematical proof is done for the conclusion.
2. The model performs well compared with other unsupervised baselines.
3. The proposed framework may be applied in various models in the future.

**Weaknesses:**

1. Human evaluation might be necessary to prove the framework actually reaches the original expectation. The automatic metrics may not be adequate.
2. The MATTE does not perform well enough in the experiments. It performs poorer in accuracy and perlexity than beta-VAE. The higher scores based on word-overlap can mean little in such situation.
3. Performance of VAE-based models always varies according to random factors, and it’s better to note the mean and std. of the results in multiple trials.

**Questions:**

Thanks for authors' detailed rebuttal, which addressed my concerns. I'll keep my score.

**Limitations:**

Limitation discussed in the paper.

---

> ### Author Rebuttal · Authors · 2023-08-10
>
> Thank you so much for the insightful comments and suggestions.  We hope that generation tasks, including advanced LLMs would benefit from principles for representation learning, as developed in our work.
>
> ### Human Evaluation
> Thank you for providing us with valuable suggestions to enhance our work. In line with the settings in [10], we have established the following criteria to assess the style, meaning, fluency, and overall transfer quality. Due to time constraints, we randomly selected 100 examples from the four datasets and gathered the generated sentences from the top-performing baselines in each group, namely CPVAE, Optimus, and our approach. We have invited three English-fluent evaluators to rate the first three metrics using a 5-point Likert scale (higher scores indicating better performance) and ask them to rank the three generated sentences (tied ranks are permitted). The Cohen’s Kappa coefficient among every two annotators over the rank-based metric 0.46. The results are supplied in **Table 1** in the attached PDF.  The results show that human annotators favour Optimus in terms of content preservation and fluency, which can be attributed to its powerful decoder in generating fluent sentences. After considering the style transfer correctness, Matte ranks as the best-performing method with more than 58% support set.
>
> ### $\beta$-VAE performance
>
> Thanks for pointing out this result. $\beta$-VAE indeed demonstrates higher acc and lower perplexity. However, we have observed that many generated sentences follow simple but repetitive patterns. For example, 2.2\% transferred sentences in the Yelp domain containing the phrase “I highly recommend” while only 0.6\% original sentences do. While these sentences are fluent and express the desirable sentiment, they significantly differ from the original sentences and indicate a generation degradation problem.
>
>  In this context, BLEU can play a role in avoiding pitfalls because the significantly lower BLEU probably indicates the decline in the model reconstruction ability-language modelling ability. Therefore, we use G-score as the primary metric to balance sentiment acc and semantic preservation.
>
> We also noticed another metric, “diversity-n”[9] can be a good indicator for oversimplified and repetition generation. It measures the ratio of distinct n-grams in all the n-grams in the generated sentences. All the other methods except for $\beta$-VAE generated sentences with similar diversity-2 as the original sentences, but the sentences generated by $\beta$-VAE have much lower diversity than the original ones (see Table below).  We will make it clearer to the audience. Thank you for your valuable comments again.
>
> **Table**: Diversity-2 for the transferred sentences. Diversity for the original sentences is added in the bracket for comparison. $\beta$-VAE has significantly fewer distinct 2-gram than original datasets.
> | Dataset     | IMDB (0.34) | Yelp(0.63) | Amazon(0.64) | Yahoo(0.44) |
> |-------------|-------------|------------|--------------|-------------|
> | $\beta$-VAE | 0.11        | 0.46       | 0.37         | 0.22        |
> | JointTrain  | 0.21        | 0.59       | 0.56         | 0.37        |
> | CPVAE       | 0.32        | 0.59       | 0.57         | 0.45        |
> | MATTE       | 0.32        | 0.62       | 0.61         | 0.45        |
>
> ### BLEU metric
>
> Thank you for sharing your insights. Although many existing papers adopt BLEU [1, 2, 3, 4, 5] and it is still an open question of how to measure semantics preservation due to word variation [6], we completely agree with you that BLEU has limitations in capturing semantic relatedness beyond literal word-level overlap. To make this issue clearer to the audience, we will make a footnote in the paper. We can observe that another metric, sentiment accuracy, also favours our methods similarly. Furthermore, we adopt one metric, CTC score [7], to avoid the aforementioned issues of BLEU, as it considers the matching embeddings, i.e., cosine similarity of pretrained word embedding rather than the “hard match”. The results in **Table 4** in the attached PDF show $\beta$-VAE display the least impressive performance and, Optimus and Matte exhibit the overall best results. Admittedly, results are less discriminative than BLEU -- this phoneme is also observed in [8].
>
>
> ### Mean and std.
>
> Thank you for the suggestion! We will add the mean and std to our draft. We also included the results in **Figure 1** in the PDF.
>
> Please let us know if you have further concerns -- thank you!
>
> ### References:
>
> [1] Semi-Supervised Formality Style Transfer with Consistency Training, ACL22
>
> [2] Deep Learning for Text Style Transfer: A Survey, CL21
>
> [3] A Probabilistic Formulation Of Unsupervised Text Style Transfer. ICLR20
>
> [4] Style transformer: Unpaired text style transfer without disentangled latent representation, ACL19
>
> [5] Unsupervised text style transfer using language models as discriminators. Neurips18
>
> [6] Repairing the cracked foundation: A survey of obstacles in evaluation practices for generated text. JAIR23
>
> [7] Compression, Transduction, and Creation: A Unified Framework for Evaluating Natural Language Generation. EMNLP21
>
> [8] Composable Text Controls in Latent Space with ODEs. Arxiv 2022
>
> [9] A diversity-promoting objective function for neural conversation models. ACL16
>
> [10] A Review of Human Evaluation for Style Transfer. GEM21

---

> > ### Comment · Reviewer_1Pqp · 2023-08-15
> >
> > Thanks a lot for your clarification with details.  I wonder if you could update the paper properly within the limitation of the pages, since you give so many updates for the reviewers.

---

> > > ### Author Response · Authors · 2023-08-16
> > >
> > > Thank you for the thoughtful question. We have incorporated the indicated updates as follows to ensure that the paper is informative and meets the page limit.
> > >
> > > - We included the human evaluation results and results from the tense-transfer task as two separate tables in Section 6.1 and Section 6.3 (a new subsection).
> > > - We included a discussion on the degradation issue of $\beta$-VAE in Section 6.1 (sentiment transfer performance), and we refer the readers to Appendix A.4 for the Diversity-2 measurement table.
> > > - All results now feature both mean and std.
> > > - We included remarks on comparing our principle-based generation approach with the anatomy-replacement method and ChatGPT at the closing of Section 5, and we refer the readers to Appendix A.5 for experimental results.
> > > We included the CTC score–BLEU alternative evaluation and multiple style classifier evaluation results in Appendix A.3 and allude to them in Section 6 (Evaluation metric).
> > > - We included suggested references in Section 2 (related work).
> > > - For short text edits (e.g., typos, minor remarks, more details, and footnotes), we made adjustments to the corresponding paraphrases.
> > >
> > > To abide by the page limit, we shortened and merged the two “contrast with prior work” paragraphs in Section 4.1 and Section 4.2 and placed the abridged version by the end of Section 4. We shortened the baseline description in Section 6.1 and deferred the detailed version to the Appendix. We condensed texts in Sections 1 (introduction) and 7 (conclusion).
> > >
> > > As uploading drafts is not permitted at this stage, we share a few revised paragraphs below.
> > >
> > > The discussion on the comparison with anatomy-replacement methods and ChatGPT (now in Section 5):
> > >
> > > >As well acknowledged, large language models (LLMs) have demonstrated an impressive ability to generate fluent text. That said, we view the principles for counterfactual generation as complementary to the development of LLMs, and we hope that our theoretical insights can further enhance LLMs. We supply examples in Table 11 (Appendix) that LLMs falter on sentiment transfer for overlooking overall and implicit sentiments, although they can precisely replace sentiment-related words. Thus, one may anticipate that LLMs would benefit from principles for representation learning, as developed in our work.
> > >
> > > Contrast with prior work (previously Section 4.1 and 4.2, now merged and placed in Section 4.2):
> > >
> > > >Zheng et al. enforce absolute sparsity constraints on latent component influences. In comparison, Theorem 1 requires relative sparsity between two subspaces, which could be better suited for language-related applications. Unlike Kong et al., which assumes subspace independence, our method acknowledges the interdependence between the two subspaces, a common scenario in language contexts. Assumption 2 enforces separate influences for content and style to facilitate style identification. Conversely, Kong et al. hinge their proof on content-style independence, limiting its applicability here. We refer the readers to Appendix A.8 for a detailed comparison.
> > >
> > > The degradation of $\beta$-VAE issue is added in Section 6, sentiment transfer performance:
> > >
> > > >Among LSTM baselines, $\beta$-VAE shows high sentiment transfer accuracy and fluency but poor content preservation. We have observed that many generated sentences follow simple but repetitive patterns, e.g., 2.2% transferred sentences in Yelp containing the phrase “I highly recommend” while only 0.6% original sentences do. These sentences are fluent and express the desirable sentiment but significantly differ from the original sentences, indicating generation degradation.~\footnote{The metric diversity-n (Li et al.) can also indicate repetitive generation pattern as it measures the ratio of distinct n-grams in all the n-grams in the generated sentences. We add the complete evaluation results in Table 9 (Appendix).}
> > >
> > > Thank you for your constructive comments, and please let us know if you have any suggestions – many thanks!

---

### Official Review · Reviewer_zc3n · 2023-07-06

**Soundness:** 2 fair
**Presentation:** 2 fair
**Contribution:** 2 fair
**Rating:** 4
**Confidence:** 3

**Summary:**

This paper wants to do a controllable text style transfer by tackling the dependence between the content and the style variables. They adopt the concept of influence sparsity requiring the influence of the style variable to be sparser than the content variable. They evaluate their method on several NLP datasets to show the style transferred text generation.


**Strengths:**

The paper aims to do controllable text style transfer which is an interesting application in NLP domain. In this paper the authors relax the independence between the style and context fills the gap in the literature.

**Weaknesses:**

The main idea of this paper is to disentangle the content from the style. However throughout the whole paper, I don’t think the authors define clearly what is a “style”. In the literature, people usually define the sentence structure (e.g, dependency parsing tree) as the “style” and the semantics as the “content”. While in this paper, in Introduction (line 37), the author mentioned a positive sentiment is “style”, but later in Sec.3 (Line 113), they refer to the positive descriptions of something as “content”.

The experiment only shows the style transfer on sentiment perspective and lacks comparison to many advanced baselines.


**Questions:**

- The author mentioned their method could leverage the data from multiple domain. How is this reflected in your results?
- In your experiment, the goal is to do a “style” transfer which means transferring the sentiment. However the accuracy (which means the intended attributes are expressed, Line 299) is very low, how can you guarantee your style transfer is successful? Also what is the label distribution for each data? What is the accuracy of a random guess in each case?
- From Table 2, the transferred texts seem only change several tokens. So how does your method compare to a naive antonym replacement?
- There are other unsupervised text-style transfer learning algorithms that are more up-to-date, please check: https://github.com/fuzhenxin/Style-Transfer-in-Text

---

> ### Author Rebuttal · Authors · 2023-08-10
>
> Thank you so much for the valuable comments and suggestions.  We address each point as follows.
>
> ### Advanced baselines and evaluation on other styles
>
> Thanks for providing the comprehensive list! It should be noted that our proposed method does not rely on attribute/style labels in the training, nor the paired data, while the unsupervised methods in the list are only free from the paired data. They still rely on the style annotation for each training sample. In this sense, our method is similar to unsupervised disentangled learning ($\beta$-VAE, DAAE[1], DCTC [2]). As far as we know, the most recent work without style labelling should be CPVAE [3], and both CPVAE and our Matte only need a small set of sentences with the desired styles to derive $\tilde{s}_\text{transfer}$ when conducting the sentiment transfer (inference phase). See Sec5.3.
>
> Our main contribution is to propose two flexible assumptions to replace existing unrealistic assumptions.  It can be applied to broader generation problems -- the sentiment transfer is one example that satisfies our assumption that sentiment is relatively sparse and is influenced by the content.  Thanks for your comments about experiments on more styles, we add an experiment on tense transfer, which transfers the tense between present and past. Specially, we follow [1] to extract the tense for the main verb using the StanfordNLP Parser, and measure the tense of the transferred sentences by the parser. We compare with DAAE and DCTC and the results are in **Table 3** in the PDF. Our method outperforms the other baselines over 6\%, with an illustration of a discriminative representation s for different tense.
>
> ### Style or Content
>
> Thanks for raising this fundamental question. We agree that the usage of 'style' might cause confusion, as you pointed out. In this paper, we decompose latent variables into two parts, which follow the assumed generating process to facilitate counterfactual generation. It uses a broader sense of "style", which are latent variables that have a sparse influence on the text (i.e., relative to the "content" latent variable) and are influenced by specific content variables. We believe that sentiment, in many cases, falls into such a category, as discussed in the examples, e.g., lines 36-39. Thanks to your comment, we will consider renaming the two latent variables to avoid confusion cause by “content” and “style” or add a footnote to make this point clear to readers.
>
> ### Domain in results
>
> Thank you for the nice question! We train all baselines and the proposed approach with multiple domain datasets. They differ in how to leverage the domain information.  The effects can be seen from both quantity and qualitative aspects:
> In Table 1, all the baselines in the LSTM group first derive a domain embedding $u$ and then concatenate $u$ to the sentence-level representation to differentiate their domain source. For our approach built on CPVAE, we adopt $u$ in the content-style dependency modelling described in Eqn(2). We can see clear improvement over CPVAE. Besides, in Table 2, the first example, “The guy is an awful actor → The guy is very flavour” shows that the simple concatenation of domain indexes in CPVAE fails to capture the domain-adaptive content-style dependency: the sentiment change leads to unnatural content-style dependence, although the polarity is correctly flipped.
>
> ### Analysis of style acc results
>
> We apply a sentiment classifier with 95% acc in all the validation datasets to evaluate whether the intended sentiment is expressed in the transferred sentences. That is, if the original sentence is positive/negative (according to the data annotation) and the transferred sentence is negative/positive (evaluated by the classifier), then the sentiment accuracy numerator is increased by one. So, it is not applicable that the sentiment acc has to be over 50% (random guess) if there is only binary sentiment polarity.
>
> ### Antonym replacement baseline
>
> Thanks for sharing your insights. We totally agree that word substitution is important in many text rewriting tasks, and naive antonym replacement can flip the overall sentiment with minor modifications when the given sentences have simple structures and explicit sentiment indications. We supply the comparison results in the table below with antonym replacement over the four datasets. Specially, we adopt the NLTK tool to do pos tagging to the original sentence and replace the words annotated as “ADJ” (adjective) with its antonym found in WordNet.  The performances vary a lot from different datasets. Our method performs relatively stable across different datasets. It can be partly explained by the variations of the sentences and the representation learning approach can benefit generalizability. Some examples are shown below. We will add a footnote to Table 2 to elaborate on this insight.
>
> **Table**:  Sentiment acc of antonym replacement and Matte.
>
> |                     | Imdb  | Yelp  | Amazon | Yahoo |
> |---------------------|-------|-------|--------|-------|
> | Antonym replacement | 11.40 | 14.40 | 10.30  | 21.75 |
> | Matte               | 32.43 | 34.30 | 34.50  | 38.45 |
>
> We also supply some cases below:
> ~~~
> src: Need a cheap car charger and it seems to do the job.
> anto: Need a expensive car charger and it seems to do the job .
> ours: Need a cheap car charger and it seems to be unaffordable.
>
> src: I will stick to my Xbox for now, thanks.
> anoto: I will stick to my Xbox for now, thanks.
> ours: I will buy it to replace my Xbox, thanks.
> ~~~
>
> Please let us know if you have further concerns, and please consider raising the score if we have addressed your concerns -- thank you!
>
> ### References:
>
> [1] Educating Text Autoencoders: Latent Representation Guidance via Denoising. ICML20
>
> [2] Disentangling Generative Factors in Natural Language with Discrete Variational Autoencoders. EMNLP21
>
> [3] On variational learning of controllable representations for text without supervision. ICML20

---

> > ### Author Response · Authors · 2023-08-18
> >
> > We've taken your initial feedback into careful consideration and incorporated them into our manuscript as indicated in our response. Could you kindly confirm whether our responses have appropriately addressed your concerns? If you find that we have properly addressed your concerns, we kindly request that you consider adjusting your initial score accordingly. Please let us know if you have further comments.
> >
> > Thank you for your time and effort in reviewing our work.

---

> > ### Comment · Reviewer_zc3n · 2023-08-20
> >
> > Thanks for the rebuttal. However, after reading other reviews as well, I feel there are quite some efforts needed for the paper to improve and solve all the reviewers' concern. I will keep my original score.

---

> > ### Author Response · Authors · 2023-08-20
> >
> > Thanks for your follow-up insights!  As indicated in our responses to other reviewers, we have responded to all the explicit concerns raised by all the reviewers and updated our manuscript accordingly.
> >
> > As uploading drafts is not permitted at this stage, we share some revised paragraph examples below:
> >
> > To highlight the **effects of domain $u$ in our results**, we modified the paragraph **sentiment transfer performance** in **Section 6.1**.
> > >Our model is built on top of CPVAE with the proposed causal influence modules and
> > sparsity regularisations.  Specially, we adopt $u$ to establish the domain-varied dependency between content and style, illustrated in Eqn(2), while all the baselines in the LSTM group first derive a domain embedding and then concatenate it to the sentence-level representation to differentiate their domain sources. We can see clear improvements over the best-performing baseline CPVAE, which can be partly explained by our novel method of incorporating domain information.  Moreover, the first example "*The guy is very flavour*" in Table 2 shows that the simple concatenation of domain index in CPVAE fails to capture the domain-adaptive content-style dependency and leads to an unnatural content-style match.
> >
> > We included remarks on comparing our principle-based generation approach with the **anatomy-replacement method and ChatGPT at the closing of Section 5**, and we refer the readers to **Appendix A.5, Table 11 and Table 12** for experimental results.
> > > **Comparison with large language model**. As well acknowledged, large language models (LLMs) have demonstrated an impressive ability to generate fluent text. That said, we view the principles for counterfactual generation as complementary to the development of LLMs, and we hope that our theoretical insights can further enhance LLMs. We supply examples in Appendix, Table 11 that LLMs falter on sentiment transfer for overlooking overall and implicit sentiments, although it can precisely replace sentiment-related words. Thus, one may anticipate that LLMs would benefit from principles for representation learning, as developed in our work~\footnote{We also provide the results of anatomy replacement in Appendix, Table 12, to further compare the interventions that occur in input-space and latent-space.}.
> >
> > We included the results of the tense transfer task in **Section 6.3 (a new subsection)**, with newly added **Table 5** and **Figure 6** as illustrations.
> > >To further verify our theoretical insights, we apply MATTE to tense transfer between past and present, in which tense is the style variable with relatively sparse influence to sentence. We reuse the model trained on the above four datasets and do inference on the Yelp dataset. Specially, we follow Shen et al. to identify the tense of sentences by extracting the main verb using the StanfordNLP Parser. In order to transfer the tense from past to present, we collect 100 present sentences in the dev set to derive $s_\text{transfer}$ as a replacement of the original $s$ for the past sentences, and vice versa.
> >
> > >The tense transfer accuracy results on Yelp test tet (Table 5) show significant improvements, 6% from MATTE over the best-performing baseline. We compare the learnt style variable $s$ derived from CPVAE (left) and MATTE (right) in Fig 6. There is a clear bond between past and present sentences in MATTE, while some past (blue dots) are mixed in the bottom of the red district in CPVAE, which implies that MATTE learns a better disentangled tense representation.
> >
> > We hope the updated text could address your concerns. Please kindly let us know if you have further concrete questions or concerns. Thank you for engaging in our work!

---

> > > ### Author Response · Authors · 2023-08-21
> > >
> > > Dear reviewer zc3n,
> > >
> > > Once again, we are grateful for your time and efforts.  Since the discussion period will end in one hour, we will be online waiting for your feedback on the further response we provided yesterday. We would highly appreciate it if you could take into account our response when updating the rating and having discussions with AC and other reviewers.
> > >
> > > Thanks for your contribution to NeurIPS 2023!
> > >
> > > Authors of #1309

---

### Official Review · Reviewer_pDz1 · 2023-07-13

**Soundness:** 2 fair
**Presentation:** 3 good
**Contribution:** 2 fair
**Rating:** 6
**Confidence:** 3

**Summary:**

The work "Controllable Text Generation with Identifiability Guarantee" presents a model for unsupervised counterfactual generation based on a model with disentangled style and content. The model is based on a variational autoencoder augmented with two flow-based models operating in the latent space. The flow-based models perform the disentanglement and, since they are invertible, allow to intervene on the style variable for counterfactual generation.

**Strengths:**

Overall, I find the basic idea and the model itself interesting and novel. I like how the paper presents theoretical results that are then illustrated by the model rather than just an experimental evaluation. However, there are several reservations that weigh the scales towards rejection for me.

**Weaknesses:**

First, the experimental evaluation is underwhelming, for several reasons:

(i) the only setting provided is sentiment transfer, which significantly restricts the claims made for counterfactual generation in general; in fact, I would argue that the introduction and abstract are much more general than the actual results, and would advise to rewrite the introduction to mention that the model is only proven to work for sentiment transfer;

(ii) I'm not sure that the BLEU metric and consequently the G-score make a lot of sense here since BLEU simply shows how much of the original wording is preserved; e.g., replacing a word with a synonym reduces BLEU but, all else being equal, arguably makes for better counterfactual generation since it makes generated sentences more varied and hence useful, e.g., as synthetic data;

(iii) moreover, I'm not sure I understood the accuracy metric as presented: e.g., the IMDB dataset only has positive and negative sentiments, the authors claim that their classifier has 95% accuracy on original validation sets (this makes sense), but then Table 1 shows IMDB accuracies for counterfactuals ranging from 14% to 38% -- so that's much worse than chance for all methods including the supervised upper bound?.. it may be a misunderstanding on my part, but the paper does not clarify this at all;

(iv) the qualitative results in Table 2 are also unclear: e.g., the authors claim that Optimus alters the semantics but doesn't MATTE also do it in Src 2? plus, I couldn't understand Src 4 at all, either the original or transferred versions, they don't make any sense.

Second, I'm afraid that these days a text generation model has to compare itself with modern large language models, while the authors choose a GPT-2-based model from 2019 as their best baseline (and a baseline that serves as an unachievable upper bound since it's supervised). I wonder how well GPT-3.5 would do if you just asked it nicely to "please invert the sentiment while preserving content as much as possible in the following sentence that originates from the domain of movie reviews", maybe with a couple of generic examples in the prompt?

Third, related work would benefit from a section on topic modeling that has developed joint sentiment-topic models such as ASUM or USTM that are quite similar in their basic assumptions to the presented model. Topic models obviously cannot serve for counterfactual generation but it looks like they were an influence.

**Questions:**

Please see previous section

**Limitations:**

The approach proposed in the paper has some technical limitations leading to a quality decrease. However, the proven identifiability guarantees lead to the mitigation of some of the potential societal risks. E.g. if the algorithm is used to change the style of some text, its semantics should be kept intact, to avoid the occasional creation of misinformation messages.

---

> ### Author Rebuttal · Authors · 2023-08-10
>
> Thank you so much for the valuable comments and suggestions. We have revised and included some results and we hope the revision could address your concerns.
>
> ### Task is limited
>
> Thank you for the suggestion! We will make it explicit that our empirical study is limited to sentiment transfer.
> At the same time, to make our contribution even clearer, we will highlight that our primary contribution is to establish a theoretical foundation for counterfactual generation and reason about theoretical guarantees, which is often missing in the existing literature. The resulting theory and framework are general and can inform the algorithmic design for many other tasks.
>
> Furthermore, we evaluate MATTE on tense transfer task and results are in **Table 3** in the PDF.  Specially, we follow [5] to extract the tense for the main verb using the StanfordNLP Parser,  evaluate the transferred sentences by the parser. It outperforms the best baseline over 6\% with impressive style representation mapping results.
>
> ### BLEU is not good
>
>
> Thank you for sharing your insights. Although many existing papers adopt BLEU [1, 2], we completely agree with you that BLEU has limitations in capturing semantic relatedness beyond literal word-level overlap. To make this issue clearer to the audience, we will make a footnote in the paper. We can observe that another metric, sentiment accuracy, also favours our methods in the same way. Furthermore, we adopt one metric, CTC score [3], to avoid the aforementioned issues of BLEU, as it considers the matching embeddings, i.e., cosine similarity of pretrained word embedding rather than the “hard match”. The results in **Table 4** (in the PDF) show that $\beta$-VAE displays the least impressive performance, and Optimus and Matte exhibit the overall best results. Admittedly, results differences are less discriminative than BLEU -- this phoneme is also observed in [4].
>
> ### Sentiment acc
>
> We apologize for any confusion. The evaluation has successfully achieved a 95% accuracy for binary sentiment classification on the original sentences in the dev set, showcasing its reliability as a sentiment classifier. Consequently, we employ this classifier to assess whether the intended sentiment is accurately conveyed. Specifically, if the transferred sentence is assessed as the opposite sentiment (evaluated by the sentiment classifier), then the sentiment accuracy numerator is incremented by one. In the worst case, all the generated sentences fail to express the opposite sentiment after the intervention, and the sentiment acc is 0. Therefore, the random sentiment transfer acc is not 50\% as in classification problems.
>
> ### Case study examples are not clear
>
> Thanks for your valuable comments on the qualitative results. We will improve this part by providing clearer examples and descriptions in the manuscript.
>
> ###  Compared to GPT3.5
>
> We fully agree with you that evaluating LLMs will provide valuable insights. As well acknowledged, LLMs have demonstrated an impressive ability to generate fluent text. That said, we view the principles for counterfactual generation as complementary to the development of LLMs, and we hope that our theoretical insights can further enhance LLMs. In fact, we supply examples below that LLMs falter on sentiment transfer. Chatgpt can precisely detect the sentiment words and find a good replacement to complete a fluent sentence. This word-level replacement favours sentences with explicit and straight sentiment expressions. Our methods, based on representation learning, can probably better capture mixed and implicit sentiment expression, where ChatGPT neglects the overall sentiment. Thus, one may anticipate that LLMs would benefit from principles for representation learning, as developed in our work.
>
> Examples: Source sentence (S), ChatGPT (C), MATTE (M).
> ~~~
> S: I had to leave a message, and they never called me back.
> C: I had to leave a message, and they promptly called me back.
> M: I had to leave a message, and they kept thinking that ended up that finally helped me.
>
> S: This case is cute however this is the only good thing about it.
> C: This case is not cute; however, it is the only good thing about it.
> M: This case is cute and overall a valuable product.
>
> S: The buttons to extend the arms worked exactly one time before breaking.
> C: The buttons to extend the arms failed to work from the beginning, never functioning even once.
> M: The buttons to extend the arms worked exactly as described.
> ~~~
> ### Connection with topic modelling:
>
> Thank you for the great suggestion! Taking the ASUM as an example, it is built on Latent Dirichlet Allocation (LDA) and models the generation of words based on the hierarchy of latent variables that is conditional on sentiment. Words are generated from the sentiment-aspect-word multinomial distribution. It requires a predefined list of polarity words to set the asymmetric sentiment priors. In contrast, in our model, the style (or sentiment) depends on the content (or aspect), and we don’t need to pre-define the sentiment prior.   More concretely, the key differences are (a) we introduce the domain index $u$ to allow for varying dependencies between content and style across different domains; (b) we can apply an arbitrary function to model the dependency between content and style, i.e., Eqn (2); (c) based on the generation process, we propose two flexible constraints to the distributions to achieve the identifiability guarantee.
>
> Please let us know if you have further concerns, and please consider raising the score if we have clarified your concerns -- thank you!
>
> ### References
>
> [1] Semi-Supervised Formality Style Transfer with Consistency Training
>
> [2] Deep Learning for Text Style Transfer: A Survey
>
> [3] Compression, Transduction, and Creation: A Unified Framework for Evaluating Natural Language Generation
>
> [4] Composable Text Controls in Latent Space with ODEs
>
> [5] Educating Text Autoencoders: Latent Representation Guidance via Denoising

---

> > ### Author Response · Authors · 2023-08-18
> >
> > We've taken your initial feedback into careful consideration and incorporated them into our manuscript as indicated in our response. Could you kindly confirm whether our responses have appropriately addressed your concerns? If you find that we have properly addressed your concerns, we kindly request that you consider adjusting your initial score accordingly. Please let us know if you have further comments.
> >
> > Thank you for your time and effort in reviewing our work.

---

> > > ### Author Response · Authors · 2023-08-21
> > >
> > > Dear reviewer pDz1,
> > >
> > > Once again, we are grateful for your time and efforts.  We have been eagerly waiting for your feedback on our point-to-point response.  We will be here waiting and hope to see it before the discussion period ends. We understand that you are very busy, but would highly appreciate it if you could take into account our response when updating the rating and having discussions with AC and other reviewers.
> > >
> > > Thanks for your time,
> > >
> > > Authors of #1309

---

> > ### Comment · Reviewer_pDz1 · 2023-08-21
> >
> > Thanks authors for carefully addressing my comments. New experiments provided in the rebuttal dismiss most of my concerns. I raise the score to 6.

---

> > > ### Author Response · Authors · 2023-08-21
> > >
> > > Thank you so much for providing valuable feedback and acknowledging our work! We will incorporate them carefully into the future version -- many thanks for your time and effort!

---

### Official Review · Reviewer_pLx5 · 2023-07-24

**Soundness:** 2 fair
**Presentation:** 1 poor
**Contribution:** 3 good
**Rating:** 4
**Confidence:** 3

**Summary:**

This paper discusses controllable text generation and tackles the dependence between content and style in the counterfactual generation task. Identification guarantees are proven and used to enhance disentangling of the variables.

**Strengths:**

Style and content disentanglement, especially for scenarios with sparse data, is very challenging and important to language generation.

A theoretical discussion on sparsity constraints is interesting.

The following practice is motivated by the theory and the experimental results are quite positive.

**Weaknesses:**


+ The writing requires improvement:

The definition of some key concepts of this paper are missing, such as 'identifiability guarantee' and 'relative sparsity'.

The connection between the generative model (described in Sec 3) to previous works is not discussed. Comparison to an existing counterfactual generation framework is appreciated.

The paper confounds 'sentiment' and 'style', which gets worse when it is heavily used as the running example throughout the paper. Sentiment is more of the semantic aspect of the text.

Many methods and technical choices are only expressed in math without enough explanation and motivation.

+ Some technical questions:

u is used in Sec 3 but missed in Sec 4. How is it considered?

What is the definition of T(z)?

Assumption 1.i requires g to be invertible, but sparse matrices are usually not invertible, as they are not full rank.

The connection between the proposed theory and practice is loose. One issue is the assumptions are strict, while they are used as losses (which means the assumptions may not hold during training).

Considering there are many missing definitions and explanations, I cannot judge the correctness of the theoretical part.

The latest best baseline methods [28,52] are published in 2020, you may consider later work such as [A,B] as baselines.

There is no human study and style classification only relies on one model. There could be some biases in the evaluation.


**Questions:**


+ Presentation Issues

What is 'relative sparsity'?

L21: 'state-of-art' -> 'state-of-the-art'

'grey shade' -> 'grey-shaded nodes'

Not sure about the difference between ';' and ',' in Eqn 2.

Some explanation of the assumptions using natural language would assist readers understand the work.

+ Missing reference

[A] A causal lens for controllable text generation (NeurIPS 2021)

[B] Variational Autoencoder with Disentanglement Priors for Low-Resource Task-Specific Natural Language Generation (EMNLP 2022)


**Limitations:**

The evaluation is limited to 1) only automatic methods without human inspection and 2) using a singular model architecture in experiments. All these may lead to biased discussion.

---

> ### Author Rebuttal · Authors · 2023-08-10
>
> Thank you for your valuable time and your detailed feedback! We address each point as follows.
>
> ### Writing
>
> **(1) Key Concepts.**
>
> **Identifiability guarantee** refers to the standard notion of identifiability in statistics (https://en.wikipedia.org/wiki/Identifiability), which describes the possibility of learning the true statistical model (up to certain equivalent classes) from its samples. The formal characterization of identifiability is presented in Theorem 1 (line 174) and Theorem 2 (line 216). In revision, we will highlight that identifiability refers to the standard notion in statistics. In case you see any other missing concepts, please let us know.
>
> **Relative sparsity** is formally presented as Assumption 1. iii and Figure 2. Intuitively, it prescribes that the influence from the style latent variable on the text is sparser than (thus “relative”) that from the content latent variable, as discussed in lines 188-198. This assumption encodes the belief that content information often plays a more prominent role in determining the text than style information.
>
> **(2) Comparison with existing methods.**
>
> Thanks for your constructive suggestion -- we will highlight this part to improve our paper. Our main contribution is to propose two flexible assumptions to replace existing unrealistic assumptions (e.g., the independence between $c$ and $s$) via regularisations. It can be applied to broader generation problems -- the sentiment transfer is one example that satisfies our assumption. Our framework is fully unsupervised in training -- without paired data and style labels. In contrast, [A] incorporates style classifiers and [B] is pretrained on style-labeled data. Thus, they are not directly comparable with our approach, which is similar to unsupervised disentangled learning (Line 74). Based on CPVAE, the SOTA unsupervised style transfer model, MATTE has the following merits:
>
> (i) It is driven by the identifiability guarantee that enforces style sparsity and intersection minimization. Therefore, it is theory-grounded for the content and style disentanglement.
>
> (ii) Domain variables are introduced in the data generation process, and the dependency between content and style varies from different domains. CPVAE does not consider the multiple-domain situation.
>
> **(3) Sentiment is a style?**
>
> Thanks for raising this fundamental question. We agree that the usage of 'style' might cause confusion, as you pointed out. In this paper, we decompose latent variables into two parts, following the assumed generating process to facilitate counterfactual generation. It uses a broader sense of "style" to refer to the latent variables with a sparse influence on the text and influenced by specific content variables. We believe that sentiment, in many cases, falls into such a category, e.g., gender, tense. Thanks to your comment, we will consider renaming the two latent variables to avoid confusion or adding a footnote to make this point clear to readers.
>
> ### Technical
>
> **(1) $u$ not discussed in Sec4.**
>
> Great question! Actually, even without multiple domains, the properties (graph structures & sparsities) of the generating process grant identifiability. Therefore, we didn’t discuss $u$ in Section 4. We will add a remark to clarify this point in our revision.
>
> **(2) Definition of the $T(z)$.**
>
> $T(z)$ (defined in line 164) is the transition matrix between the Jacobian of the true generating function $J_{g}$ and that of the estimated generating function $J_{\hat{g}}$.
>
> **(3) Invertibility and sparsity.**
>
> Excellent question! We note that our conditions do not constrain the absolute sparsity of the entire Jacobian matrix. Concretely. Assumption 1. iii. only requires the Jacobian matrix dimensions corresponding to the style variable to be sparser than those for the content variable, which can be true even if the Jacobian matrix is dense. Therefore, the sparsity and the invertibility assumptions are compatible.
>
> **(4) Assumptions may not hold during training.**
>
> Thank you for raising this question! We note that the assumptions are made w.r.t. the true data-generating process that generates the dataset. We implement losses to drive the estimated model to satisfy these conditions at the training optimum. It does not affect our theory whether these conditions are met during training. The decrease in corresponding loss actually indicates better identifiability.
>
> **(5) Human evaluation.**
>
> Thank you for the valuable suggestions. Inspired by [1], we established the criteria to assess the four aspects: style, meaning, fluency, and overall transfer quality. The first three metrics use a 5-point Likert scale (higher scores indicate better performance), and the last is rank-based. We randomly selected 100 examples and compared the results from the top-performing baselines in **Table 1** in PDF.
>
> **(6) Multiple style evaluators.**
>
> We totally agree that using multiple classifiers is better to avoid performance fluctuations, though much existing work neglects this issue [2,3,4].  We adopt the pretrained BertClassifier and fine-tune it for one epoch. The model with the best dev performance (96.23%) is used in our evaluation.  The results evaluated by BertClassifier are in **Table 2** in PDF.
>
> ### Presentation
>
> We use a semicolon “;” to delineate the function inputs and the function parameters, i.e., $f$( input1, …; parameter1, …), which is often adopted in the literature. In Eqn 2, we use this notation to indicate that $g_{s}$ transforms $\tilde{s}$ to $s$, and this transformation is determined by $c$ and $u$.
>
> Please let us know if you have further concerns and consider raising scores if we have clarified your concerns -- thank you!
>
> [1] A Review of Human Evaluation for Style Transfer. GEM21
>
> [2] Style Transfer from Non-Parallel Text by Cross-Alignment. Neurips17
>
> [3] Multiple-attribute text rewriting. ICLR19
>
> [4] A Probabilistic Formulation Of Unsupervised Text Style Transfer. ICLR20

---

> > ### Comment · Reviewer_pLx5 · 2023-08-16
> >
> > Thank you for your clarifications!
> >
> > After reading the responses, I feel this paper could be improved in the next round of modification, regarding (1) clarifying a series of fundamental concepts; (2) better explanations of the methodology and its assumptions; and (3) better organization and presentation of the paper.

---

> > > ### Author Response · Authors · 2023-08-17
> > >
> > > Thank you for feedback and the effort for reviewing our work. As indicated in our previous response, we have explicitly incorporated your feedback into our manuscript, including explicitly defined concepts like identifiability, a discussion on the references you pointed us to, and suggestions on evaluation.
> > >
> > > As uploading drafts is not permitted at this stage, we share a few revised paragraphs below:
> > >
> > > We explicitly defined identifiability at the opening of Section 4:
> > > >The notion of identifiability describes the possibility of learning the true statistical model (up to certain equivalent classes) from its samples [4]. The identifiability of a variable $z$ indicates that the estimated variable $\hat{z}$ contains all the information of $z$ without mixing the information of other latent variables. Formally, there exists an *invertible* mapping $ h $, s.t. $ \hat{z}  = h ( z ) $. This notion of identifiability is widely adopted in prior work [24,46,50].
> > >
> > > Suggested references [A] and [B] have been added to Section 2 (related work), shown below:
> > > > A line of work adopts pretrained language models as the encoder and the decoder in their model. For instance, one of the most popular pretrained VAE models, Optimus [30], employs BERT as its encoder, and GPT2 as its decoder. To train a latent connector between BERT and GPT2, it is pretrained on the wikitext dataset via an unsupervised reconstruction objective. On top of Optimus, the model in [A] introduces a pretrained classifier conditioned on the style labels, as well as two counterfactual objectives. [B] focus on transferring learning after pretraining on the style-annotated training dataset. Our model also leverages the latent variables to model the data-generating process and conducts style transfer in the latent space. However, unlike [A,B], we do not need style labels in the training phase.
> > >
> > > The human evaluation schema and results have been added in two parts of Section 6 (Experiments), i.e., evaluation metric and sentiment transfer performance:
> > > >Evaluation metric: We conduct both automatic and human evaluation. For human evaluation, we invited three English-fluent evaluators to rate the sentiment reverse, semantic preservation, fluency and overall transfer quality using a 5-point likert scale (higher scores indicating better performance) and rank the generated sentences from different models (tied items are permitted in rank).
> > >
> > > >Sentiment transfer performance:  Based on the automatic evaluation results, we randomly selected 100 examples from the four datasets and gathered the generated sentences from the top-performing baselines in each group, namely CPVAE, Optimus, and MATTE. The results in Table 2 (table is updated for human evaluation results) show that human annotators favour Optimus in terms of content preservation and fluency, after considering the style transfer correctness, Matte ranks the best-performing method with more than 58% support set.
> > >
> > > We were wondering whether your technical concerns had been properly addressed by our responses so far (If yes, could you please adjust the rating accordingly?). Please let us know if you have further concrete questions or concerns that we can address. Thank you for your engagement with our work.

---

> > > > ### Author Response · Authors · 2023-08-21
> > > >
> > > > Once again, we are grateful for your time and efforts.  Since the discussion period will end in one hour, we are very eager to get your feedback on our response. We understand that you are very busy, but we would highly appreciate it if you could take into account our response when updating the rating and having a discussion with AC and other reviewers.
> > > >
> > > > Thanks for your time,
> > > >
> > > > Authors of # 1309

---

### Author Rebuttal · Authors · 2023-08-10

We all thank all reviewers for their valuable feedback and dedicated time! We address the individual comments in separate response and will incorporate the reviews' suggestions in our revision.

---

### Decision · Program_Chairs · 2023-09-21

**Decision:**

Accept (poster)

**Comment:**

The paper describes a controllable text generation framework which is evaluated on one specific task, namely counterfactual generation.

The reviews raised a set of issues: This task is not properly defined in the paper and does not become clear from the abstract. That limits the accessibility of the paper. The link to style transfer and other conditional text generation tasks remain unclear. The term style is equated to sentiment in the paper, which is not correct; in fact it may be doubted if sentiment is a style. A proper definition of style is missing.

All these issues are, however, mitigated in the discussion, and I am confident that the paper will be improved sufficiently for the camera ready version.